# Sea ice variations and trends during the Common Era in the Atlantic sector of the Arctic Ocean

Ana Lúcia Lindroth Dauner[1], Frederik Schenk[2,3,4], Katherine Elizabeth Power[3,5], Maija Heikkilä[1,6]

[1]Environmental Change Research Unit (ECRU), Ecosystems and Environment Research Programme, Faculty of Biological and Environmental Sciences, University of Helsinki, Helsinki, 00014, Finland
[2]Department of Geosciences and Geography, University of Helsinki, Helsinki, 00014, Finland
[3]Bolin Centre for Climate Research, Stockholm University, Stockholm, 10691, Sweden
[4]Department of Geological Sciences, Stockholm University, Stockholm, 10691, Sweden
[5]Department of Physical Geography, Stockholm University, Stockholm, 10691, Sweden
[6]Helsinki Institute of Sustainability Science (HELSUS), University of Helsinki, Helsinki, 00014, Finland

*Correspondence to*: Ana Lúcia Lindroth Dauner (ana.lindrothdauner@helsinki.fi; anadauner@gmail.com)

**Abstract.** Sea ice is crucial in regulating the heat balance between the ocean and atmosphere and quintessential for supporting the prevailing Arctic food web. Due to limited and often local data availability back in time, the sensitivity of sea-ice proxies to long-term climate changes is not well constrained, which renders any comparison with palaeoclimate model simulations difficult. Here we compiled a set of marine sea-ice proxy records with a relatively high temporal resolution of at least 100 years covering the Common Era (past 2k) in the Greenland-North-Atlantic sector of the Arctic to explore the presence of coherent long-term trends and common low-frequent variability and compared those with transient climate model simulations. We used cluster analysis and empirical orthogonal functions to extract leading modes of sea-ice variability, which efficiently filtered out local variations and improved comparison between proxy records and model simulations. We find that a compilation of multiple proxy-based sea-ice reconstructions accurately reflects general long-term changes in sea-ice history, consistent with simulations from two transient climate models. Although sea-ice proxies have varying mechanistic relationships to sea-ice cover, typically differing in habitat or seasonal representation, the long-term trend recorded by proxy-based reconstructions showed a good agreement with summer minimum sea-ice area from the model simulations. The short-term variability was not as coherent between proxy-based reconstructions and model simulations. The leading mode of simulated sea-ice associated with the multidecadal to centennial timescale presented a relatively low explained variance and might be explained by changes in solar radiation and/or inflow of warm Atlantic waters to the Arctic Ocean. Short variations in proxy-based reconstructions, however, are mainly associated with local factors and the ecological nature of the proxies. Therefore, regional or large-scale view of sea-ice trend necessitates multiple spatially spread sea-ice proxy-based reconstructions, avoiding confusion between long-term regional trends and short-term local variability. Local-scale sea-ice studies, in turn, benefit from reconstructions from well-understood individual research sites.

**Graphical Abstract**

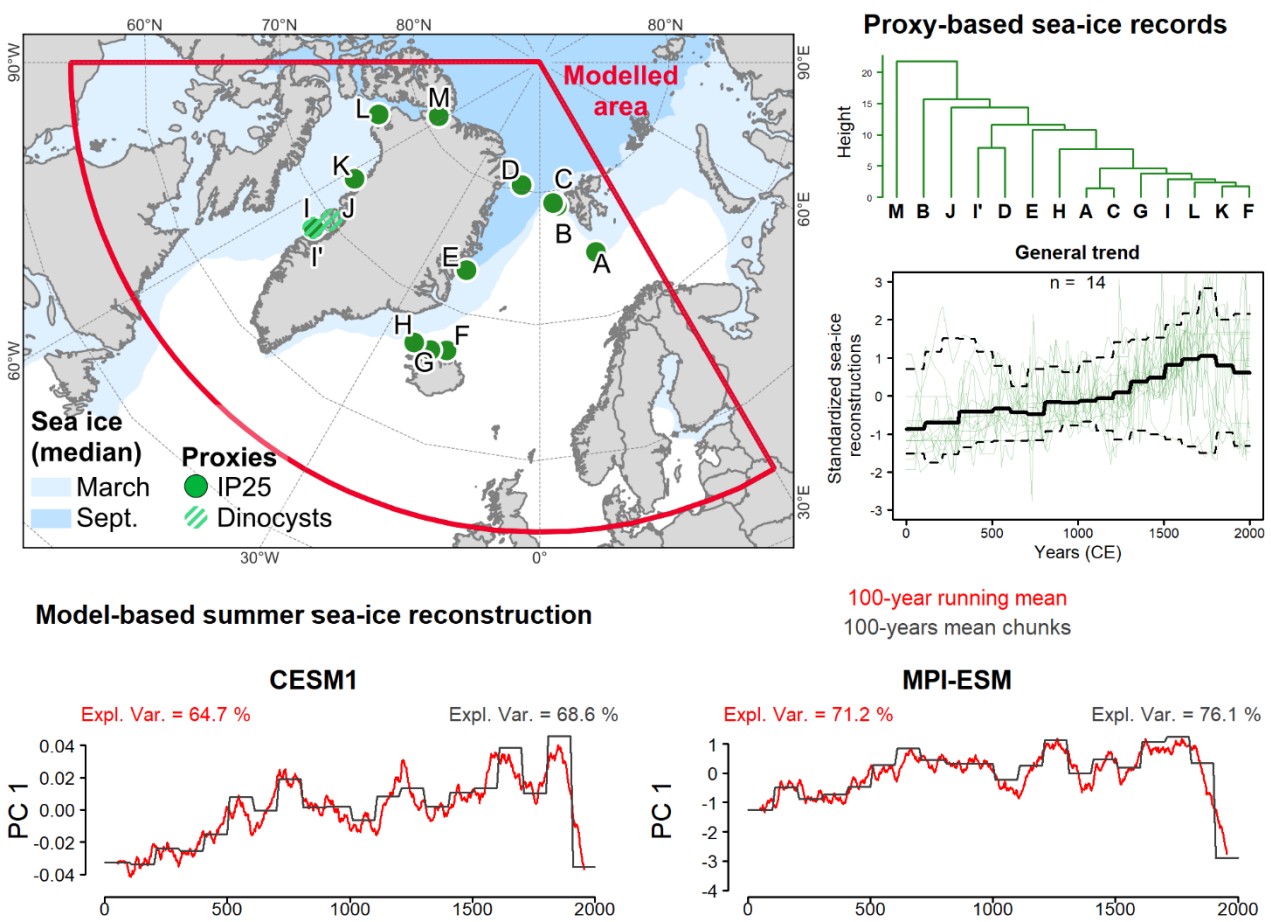

## 1 Introduction

Sea ice is crucial in regulating the heat balance between the ocean and the atmosphere and plays a major role in the Arctic food web (Arrigo, 2014; IPCC, 2021). Despite the importance of sea ice for ecosystems and climate, reconstructing its history remains challenging (Heikkilä et al., 2022). While the development of palaeoceanographic sea-ice proxies has progressed in recent decades (de Vernal et al., 2013; Belt, 2019; Weckström et al., 2020), they often exhibit strong local and regional dependencies, limiting their use for large-scale extrapolation (Brennan and Hakim, 2022). In contrast, longer transient palaeoclimate simulations of large-scale sea-ice changes using numerical models have only recently become available (Zhong et al., 2018; Mauritsen et al., 2019). Therefore, the identification and distinction between average long-term trends, and regional and internally forced variability has remained a challenge.

Sea ice affects global and regional climate systems through its role in modification of water and energy budgets (Macias-Fauria and Post, 2018; Kretschmer et al., 2020; IPCC, 2021). It acts as a physical barrier, reducing or close to preventing heat and moisture transfers between the ocean and the atmosphere, which has significant impacts on weather and climate (Bintanja, 2018; Sun et al., 2018). Also, sea ice reflects a large part of incoming solar radiation back to space due to its high surface albedo, moderating ocean warming at high latitudes (Bader et al., 2020). A reduction in high-latitude surface albedo causes a non-linear response known as Arctic amplification – a phenomenon in which northern high latitudes are warming two to four times faster than the global mean (Dai et al., 2019; Rantanen et al., 2022; Ryan et al., 2022). In addition to being a key climate parameter, sea ice presence is fundamental to Artic marine ecosystems. Sea ice provides a unique habitat for diverse biological communities formed by ice-dwelling (sympagic) and ice-dependent species (Eamer et al., 2013; Arrigo, 2014). While multiyear sea ice provides a year-round habitat that supports endemic Arctic species, areas seasonally covered by sea ice have the highest marine primary production by ice algal and phytoplankton blooms during the melt and open-water seasons (Hop and Gjøsæter, 2013; Lannuzel et al., 2020).

A major challenge for comprehending the consequences of present climate change on sea ice and Arctic marine ecosystems, is the lack of baseline data (Heikkilä et al., 2022). Although limited historical sea ice observations (mainly based on ships' logbooks) exist (Vinje, 2001; Divine and Dick, 2006; Schweiger et al., 2019), most human observations of sea ice are derived from satellites and date back only to the 1970s (Fox-Kemper et al., 2021). Palaeoclimate data derived from marine sediments can provide insights into Arctic environmental change over much longer time scales. However, most Arctic palaeoclimate data compilations so far have focused on past temperatures and the majority of the compiled data originates in terrestrial environments. For example, McKay and Kaufman (2014) compiled 56 records in an Arctic proxy temperature database for the past 2000 years, but only six record ocean temperatures. More recently, a global temperature reconstruction for the Common Era (CE) compiled more than 200 proxy records but none of the marine-related reconstructions were located in the Arctic Ocean (PAGES 2k Consortium, 2019).

To our knowledge, there are only two proxy-based data compilations of past sea-ice cover (Kinnard et al., 2011; Brennan and Hakim, 2022), neither of which focuses on marine sea-ice proxies. Kinnard et al. (2011) reconstructed sea-ice conditions

over the past 1450 years using ice-core records, tree-ring chronologies and lake sediments. Brennan and Hakim (2022) assimilated temperature-sensitive proxy records from tree rings, ice cores, corals, lake sediments and bivalve records into a numerical model to study Arctic sea-ice conditions during the Common Era. However, a wealth of evidence of past sea-ice evolution during the Common Era have been made using sea-ice proxies preserved in marine sediments, typically microfossil assemblages (e.g. Sha et al., 2017; Tsoy et al., 2017; Allan et al., 2018) and source-specific algal lipids (e.g. Belt et al., 2010; Cabedo-Sanz et al., 2016; Detlef et al., 2021). Past sea-ice reconstructions based on microfossils rely on the presence and abundance of species associated with the presence of seasonal sea ice, typically diatoms or resting stages (cysts) of dinoflagellates (de Vernal et al., 2013; Weckström et al., 2020). Highly branched isoprenoid lipids (HBIs) linked with sea-ice environments are increasingly common in sea-ice reconstructions, in particular $IP_{25}$ ("Ice Proxy with 25 carbon atoms"). It is produced by a few strictly sympagic diatoms, and has a strong link to seasonal sea-ice presence and concentration (Belt, 2019). Although the use of sea-ice proxies is becoming more widespread, local and regional forcings such as nutrient availability, freshwater and sediment contributions from land might affect the organisms whose sedimentary remains are used in sea-ice reconstructions (Brown et al., 2020; Luostarinen et al., 2020, 2023; Marret et al., 2020). Therefore, individual sea-ice reconstructions likely incorporate noise from local sea-ice characteristics and forcings other than regional-scale sea-ice cover. There are studies integrating multiple sampling sites or sediment cores, often describing and visually comparing sea-ice evolution among the records (e.g. de Vernal et al., 2013; Maffezzoli et al., 2021). However, an integration of existing sea-ice reconstructions in a systematic way for the entire northern North Atlantic to underline main trends and/or variability has, so far, not been undertaken.

Combining proxy data synthesis with climate model outputs may allow to distinguish local sea-ice changes from large-scale variability, complementing and strengthening the interpretation of past sea-ice evolution (Moffa-Sánchez et al., 2019; Brennan and Hakim, 2022). In contrast to proxies, numerical models offer improved spatial coverage and temporal resolution and can provide a coherent and physically consistent overview of the climate on varying timescales, although large model biases exist in Arctic regions, attributable to complex feedback framework and remaining deficiencies in the implementation of sea ice in general circulation models (Goosse et al., 2013, 2018). Though the majority of simulations contributing to the latest Coupled Model Intercomparison Project (CMIP6) included sea ice as a variable, detailed evaluations of sea ice in palaeoclimate simulations are still pending, as long transient simulations and more sea-ice proxies have become available only recently (Zhong et al., 2018; Mauritsen et al., 2019).

Here, we use high-resolution sea-ice reconstructions based on palaeoceanographic proxies together with numerical model outputs to identify possible common trends and/or regional differences in the evolution of sea-ice cover over the Common Era in the Atlantic sector of the Arctic Ocean. We also aim to probe whether sea-ice proxies (microfossils and $IP_{25}$) are suited to capture large-scale sea-ice changes or if local forcings have a larger influence. Lastly, we discuss the external forcings that may be causing the sea-ice changes observed in our study.

## 2 Material and Methods

### 2.1 Data

Palaeoceanographic datasets were obtained from the databases by NOAA (https://www.ncei.noaa.gov/access/paleo-search/),
PANGAEA (https://www.pangaea.de/) and Neotoma (https://www.neotomadb.org/) between January and June 2021. Additional published records that were not found in these databases were solicited from the original authors. Based on the availability of suitable records, we focused   on sediment cores retrieved from high latitudes ($> 55°$ N) and the Atlantic sector of the Arctic Ocean ($90°$ W to $30°$ E) that covered at least 80 % of the CE and presented an average temporal resolution < 100 years.

The search included data of concentrations of highly branched isoprenoid lipid $IP_{25}$ derived from sympagic algae, the abundance of diatom indicator species (*Fragilariopsis oceanica*, *F. reginae-jahniae* and *Fossulaphycus arcticus*) and the abundance of dinoflagellate cyst indicator species (*Islandinium minutum* and *I.? cezare*).

The biogeochemical proxy $IP_{25}$ was chosen due to its source-selectivity, as it is produced only by certain sympagic diatoms that are common across the Arctic Ocean (Belt, 2018). This lipid is produced in the ice during the sympagic spring bloom prior to ice melt, and is, therefore, interpreted as a proxy measure of seasonal sea ice. Since $IP_{25}$ is not produced in ice-free areas or under permanent/extensive sea-ice cover due to light limitation, changes in its concentration are interpreted in terms of fluctuations in seasonal sea-ice area (Belt, 2019).

Microfossils of sympagic diatoms are typically not found in sediments (Limoges et al., 2018; Luostarinen et al., 2023). Thus, the diatom indicator species for sea-ice are those that bloom in abundance under the ice or at the ice edge during spring melt. These species can be found in small concentrations within the ice matrix, indicating blooms seed in the ice matrix, while their main occurrence is in the stratified waters in the marginal ice zone. *Fossulaphycus arcticus*, *F. reginae- jahniae* and *F. oceanica* present a significant relationship with high April sea-ice concentrations in the Atlantic Arctic (Weckström et al., 2020).

The dinoflagellate cyst indicator species *Islandinium minutum* and *I.? cezare* were also chosen due to their affinity with seasonally sea-ice covered regions. They are found most abundantly in surface sediments in Arctic areas that experience seasonal sea ice and are mainly associated with cold waters from polar to subpolar regions (Head et al., 2001; de Vernal et al., 2020). Their ecological connection to sea-ice cover is, however, unresolved. Recently, for example, Luostarinen et al. (2023) observed that these species were virtually absent in sea ice, water column and sediment flux during spring melt although being common in surface sediments.

Concentration data were obtained in the form of mass of $IP_{25}$ either related to sediment dry weight (µg/g dw or ng/g dw) or normalized by total organic carbon (µg/g TOC or ng/g TOC). When both options were available, we prioritized $IP_{25}$ concentrations normalized by TOC. Diatom and dinoflagellate cyst abundance data were retrieved as indicator species proportion (%) of the total species assemblage. We decided to focus only on measured proxy concentrations and abundances instead of quantitatively estimates of reconstructed sea-ice concentrations and sea-ice duration. This is because quantitative

reconstructions incorporate species assemblages, including taxa not related to sea ice and primarily controlled by other environmental factors. Furthermore, the mechanistic relationships of many of the proxies to the reconstructed environmental parameter are still not thoroughly understood (Heikkilä et al., 2022). Importantly, while each proxy record used here is based on measured quantities, the proxies represent a relative measure of seasonal sea-ice concentrations rather than geophysical quantities.

The chronologies were based on radiocarbon dating (AMS $^{14}$C), and the original age-depth models were used.

## 2.2 Numerical models

Two climate simulations covering the past 2000 years (2k) were used. These are the Max Planck Institute Earth system model 1.2 Low Resolution (MPI-ESM1.2-LR; hereafter MPI-ESM) (Mauritsen et al., 2019) which was used to perform past 2k transient simulations (Jungclaus et al., 2017; van Dijk et al., 2022), and the Community Earth System Model 1.1 (CESM1.1/CCSM4; hereafter CESM1) past2k transient simulation (Zhong et al., 2018). Both models contributed to the PMIP (Paleoclimate Model Intercomparison Project) part of the Coupled Model Intercomparison Project Phase 6 (CMIP6) (Jungclaus et al., 2017).

The MPI-ESM consists of four model components: ocean dynamical model MPIOM1.6, ocean biogeochemistry model HAMOCC6, atmosphere model ECHAM6.3 and land model JSBACH3.2. These are coupled together via the OASIS3-MCT coupler. The configuration used is ~150 km in the ocean and ~200 km in the atmosphere, with 47 atmospheric vertical levels (Mauritsen et al., 2019). CESM1 is also a global fully coupled model consisting of 4 component models: atmospheric model (AGCM), ocean model (OGCM), land surface model (CLM) and sea ice model (CICE) connected by a central coupler (Craig et al., 2012). It has a ~1° resolution in the ocean and sea ice and ~2° resolution in the atmosphere (Zhong et al., 2018). Both simulations are transient, meaning they start from past climate boundary conditions, but the exact starting time varies. They then simulate the climate forward through the past2k. Forcing data for the past2k experiments is based on the PMIP4 protocol (Jungclaus et al., 2017). For CESM1, forcings are solar irradiance and volcanic aerosols (Toohey et al., 2016), land use and land cover (Klein Goldewijk et al., 2010), and greenhouse gas levels (MacFarling Meure et al., 2006). For both simulations, orbital forcing is from Berger (1978). For MPI-ESM, forcings include solar irradiance (Krivova et al., 2011), land use (Hurtt et al., 2011) and greenhouse gas levels (Brovkin et al., 2019). The only variable studied is sea ice fraction, specifically "sea-ice area percentage" (Siconca) from MPI-ESM and "fraction of surface area covered by sea ice" (Icefrac) from CESM1. These represent the percentage of ice cover in each grid square and both variables are on atmospheric grids. Extra-tropical (30 – 60 °N) surface air temperature is used in addition for model comparison (Figure 4).

## 2.3 Statistical analyses

Maps were made using QGIS (QGIS Development Team, 2022). All statistical analyses were performed in R environment (R Core Team, 2022).

Prior to statistical analyses, all proxy datasets were standardized to allow a direct comparison among data from disparate sources. Z-scores were calculated by subtracting the data by the mean and dividing by the standard deviation of each record, considering only data dated to the Common Era. To allow statistical comparisons, all datasets were reduced to a common resolution by binning to the same 100-year time slices (the top sample, centred at 1950 AD, integrated the time slice 1900 –
2000 AD, and so on) (Shuman et al., 2018).

A cluster analysis using SAX representation (Lin et al., 2007) in the dissimilarity measure calculation was used to categorize the marine proxy records into groups according to their evolution through time. SAX representation is a "structured-based" dissimilarity measure that focuses on comparing underlying dependence structures and is thus suitable for time-series analyses. The symbolization approach involves transformation of time series into sequences of discretized symbols, allowing
dimensionality reduction but keeping the distance measures defined on the original time series (Lin et al., 2007). Thus, this approach can be used to group different proxy records according to their temporal evolution, regardless of their units. For example, sea-ice reconstructions with similar patterns, such as a continuous increase, would be grouped together, separated from sea-ice reconstructions with a continuous decrease pattern. The cluster analysis was repeated using the globally (over all used records) detrended records to remove the overall dominant influence of the global trend. For this, we calculate the
average of each time step of the $z$-score records (the global trend) and subtracted this average from each of the $z$-score records. Finally, the data pertaining to each group were averaged, in order to create a unique record for each group. The TSclust package (Montero and Vilar, 2014) was used to run the cluster analyses and the zoo package (Zeileis et al., 2022) was used to deal with the missing values.

For the exploration of the climate models, we used annual sea-ice minimum and maximum of monthly mean data. Sea-ice
areas ($km^2$) were calculated by multiplying the sea ice fraction in each grid (%) by its grid area ($km^2$) and summing the area of all grids which gives the annual area. For better visualization, the data were smoothed with a 100-year running mean. To mimic the behaviour of our proxy records, we repeated the analysis using 100-years mean chunks e.g., for the EOF analysis. The "mean chunks" are meant to highlight that non-overlapping mean segments are used mimicking the samples of proxy records. Pearson correlation analysis were calculated between sea-ice areas from the whole Arctic Ocean and from the region
around Greenland using the package Hmisc (Harrell Jr., 2022).

All model data was filtered using a low-pass red noise Butterworth filter to remove sub-decadal (< 10 years) oscillations using the signal package (Ligges et al., 2022). A wavelet analysis (Morlet wavelet) was performed using the sea-ice areas (annual resolution) to identify significant oscillations that could show coherence between different datasets or represent unforced variations, using the WaveletComp package (Roesch and Schmidbauer, 2018). During the analysis, the data was
internally detrended using a degree of time series loess smoothing of 0.75, a total of 100 simulations were performed, and the used method to generate surrogate time series was a first order autoregressive model (AR-1). Wavelet analyses were also performed using the proxy data, using the same parameters. As proxy data does not have the resolution to analyse sub-decadal variability, high-frequent variability was filtered out in model runs with a low-pass filter to be consistent with proxy data.

Additionally for the model data, we applied an empirical orthogonal function (EOF) analysis for the unfiltered sea-ice fraction data to identify the main modes of variability and their spatial representation. Prior to EOF analysis, we applied a log-transformation of the fractional data using log10 after adding an arbitrary offset of 0.5 to avoid zeroes, which would be a problem for the log-transformation. The log-transformation helps to avoid the sharp cut-off at 0 and 1 that does not exist for the other variables we compare to (both proxies and simulated temperature etc.). It also helps to normalize the data

distribution and reduces skewness and heteroscedasticity to make it comparable with non-fractional data. In order to test the applicability of the 100-year time slices used for the proxy data, we calculated the EOFs using annual data, the 100-year running mean and the 100-year mean chunks. They all presented similar results, considering both EOF1/2 and PC1/2, and in this study we are presenting EOFs and PCs based on the smoothed 100-year running mean data and also the PCs based on the 100-year time-slices. EOFs were calculated with CDO version 1.9.10 (Schulzweida, 2021).

Other packages used during the analyses were dplyr (Wickham et al., 2022) to clean and organize the data, and ncdf4 (Pierce, 2015) to import the files produced by CDO into R.

## 3 Results and Discussion

### 3.1 Proxy-based sea-ice reconstructions

    Based on our spatial, temporal and resolution criteria, we retrieved 14 sea-ice reconstructions from 13 different sites (Figure

1, Table 1). Twelve of them were based on $IP_{25}$ concentrations and two were based on dinocyst assemblages. Unfortunately, there were no sea-ice records based on diatoms fitting our criteria of sufficient time coverage and resolution, and geographical boundaries.

    When dealing with individual sedimentary records, increases in $IP_{25}$ concentrations and in dinoflagellate indicator species are associated with regional increases in the presence of seasonal sea-ice cover (de Vernal and Marret, 2007; Belt and

Müller, 2013). Similarly, in this case, we assumed that increases observed in the temporal reconstruction for each cluster group refer to increases in sea-ice cover. For the temporal reconstruction of the cluster groups to present an increase in the $z$-scores, this increase must have been observed in the individual records, scattered in different regions of the studied area. Because most of the cores are located in regions that have not been covered, during the Common Era, by multiyear sea ice, an increase in seasonal sea ice in different areas can be interpreted as an overall increase in sea-ice cover. Also because of

the lack of perennial ice cover on most regions, the unimodal response of $IP_{25}$ to sea ice was discussed in detail only in the regions where it might be relevant.

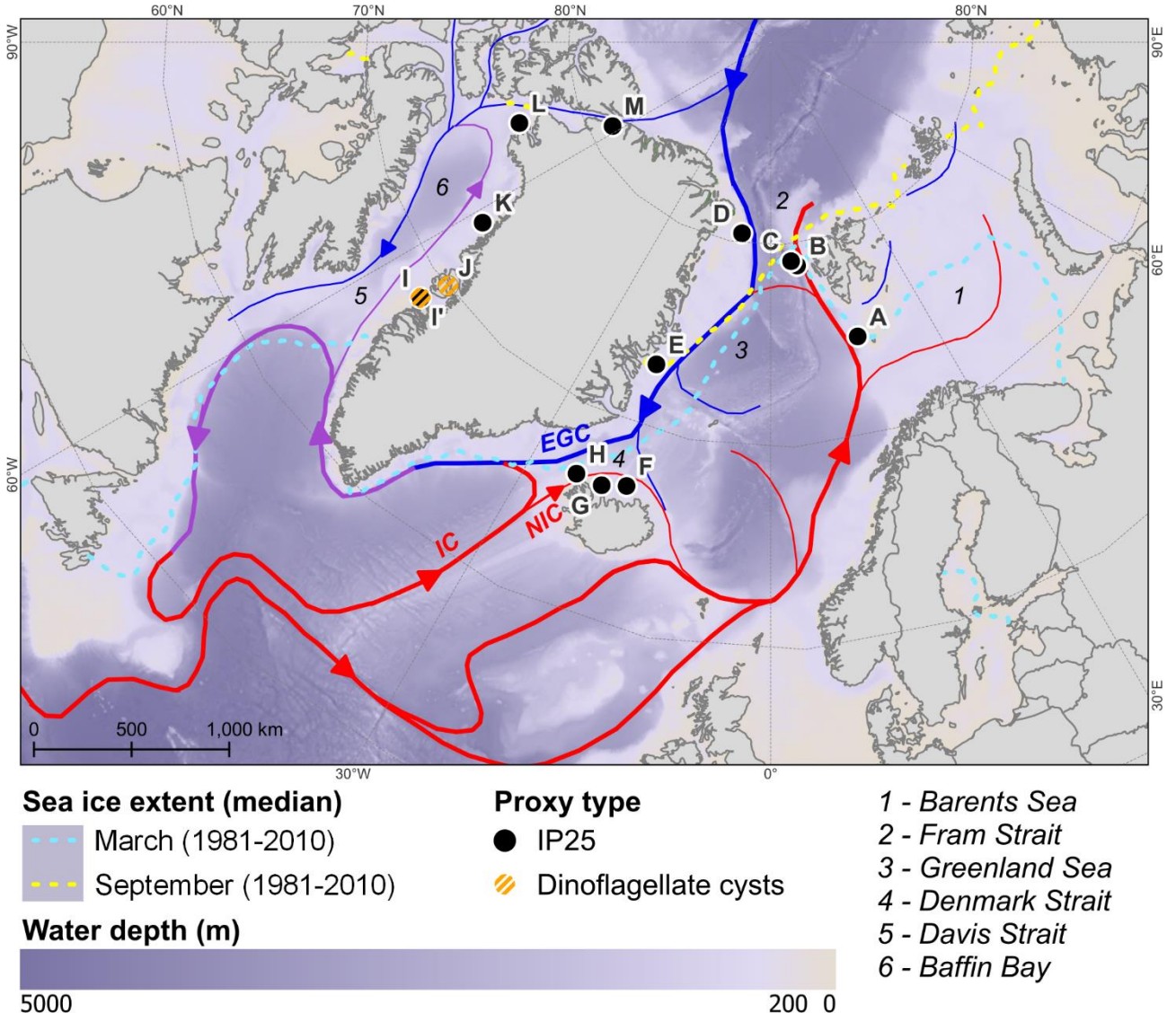

**Figure 1. Map showing the locations of IP$_{25}$ and dinoflagellate cyst records for sea-ice reconstructions. The major surface currents are shown (blue: cold waters; purple: mixed water; red: warm waters), but only the currents discussed in the text are identified. EGC: East Greenland current; IC: Irminger current; NIC: North Irminger Current. The bold black line indicates the area used for the numerical model calculations. The bathymetry was obtained from Amante and Eakins (2009), the currents were plotted based on Talley et al. (2011a) and Talley et al. (2011b), and the median sea-ice extent was obtained from NOAA-NSIDC.**

**Table 1. Marine record data including latitude, longitude, water depth and analysed variables.**

| Record | Core | Region | Latitude | Longitude | Water Depth (m) | Variable | Reference |
|--------|------|--------|----------|-----------|-----------------|----------|-----------|
| **A** | JM09-KA11-GC | Barents Sea | 74.87482 | 16.48482 | 345 | $IP_{25}$ | Köseoğlu et al., 2018 |
| **B** | MSM05/5_712-1 | Fram Strait | 78.91567 | 6.76733 | 1490 | $IP_{25}$ | Cabedo-Sanz and Belt, 2016 |
| **C** | MSM05/5_723-2 | Fram Strait | 79.16100 | 5.33783 | 1349 | $IP_{25}$ | Müller et al., 2012 |
| **D** | PS93/025-1/2 | Fram Strait | 80.48117 | -8.48867 | 290 | $IP_{25}$ | Syring et al., 2020 |
| **E** | PS2641-4/5 | Greenland Sea | 73.15500 | -19.48500 | 469 | $IP_{25}$ | Kolling et al., 2017 |
| **F** | JR51-GC35 | Denmark Strait | 66.99933 | -17.96100 | 420 | $IP_{25}$ | Cabedo-Sanz et al., 2016 |
| **G** | MD99-2269 | Denmark Strait | 66.62550 | -20.85267 | 365 | $IP_{25}$ | Cabedo-Sanz et al., 2016 |
| **H** | MD99-2263 | Denmark Strait | 66.67900 | -24.19600 | 235 | $IP_{25}$ | Andrews et al., 2009 |
| **I** | MSM05/3_343310-5-1 | Baffin Bay | 68.64768 | -53.82488 | 855 | $IP_{25}$ | Kolling et al., 2018 |
| **I'** | MSM05/3_343310-5-1 | Baffin Bay | 68.64768 | -53.82488 | 855 | dinocyst | Allan et al., 2018 |
| **J** | DA06-139G | Baffin Bay | 70.09143 | -52.89308 | 384 | dinocyst | Andresen et al., 2011 |
| **K** | AMD14-204_CASQ | Baffin Bay | 73.26105 | -57.89978 | 987 | $IP_{25}$ | Limoges et al., 2020 |
| **L** | AMD16-117Q | Baffin Bay | 77.00483 | -72.13867 | 964 | $IP_{25}$ | Jackson et al., 2021 |
| **M** | OD1507 (composite 03TC-41GC-03PC) | Arctic Ocean | 81.19133 | -62.03767 | 976 | $IP_{25}$ | Detlef et al., 2021 |


All sea-ice proxy records were initially clustered in the same group (Figure 2a), indicating that the 14 sea-ice reconstructions shared a similar overall pattern. This evidences that all records were under the influence of the same dominant forcing, producing a common trend (Figure 2b). This overall increase in sea-ice cover agrees with the reconstructed Neoglacial cooling in the Arctic (McKay and Kaufman, 2014) and globally (PAGES 2k Consortium, 2019), especially between 1200

and 1600 CE (Figure 2b, Supp. Table S1). The cooling trend and sea-ice expansion has been largely explained by the decrease in the Northern Hemisphere insolation due to orbital changes (Wanner et al., 2011) (Figure 2c). In addition, the cumulative effect of large volcanic eruptions may have contributed to this long-term cooling (Büntgen et al., 2020) culminating in the Little Ice Age (LIA; 1600 – 1850 CE in the Arctic; Wang et al., 2022) (Figure 2d). A recovery from the LIA starting around 1850 CE marks a remarkable change in the sign of trends (Figure 2b). More recently, Arctic warming

and sea-ice decrease have been accelerated by anthropogenic greenhouse gas emissions and land use changes (IPCC, 2021).

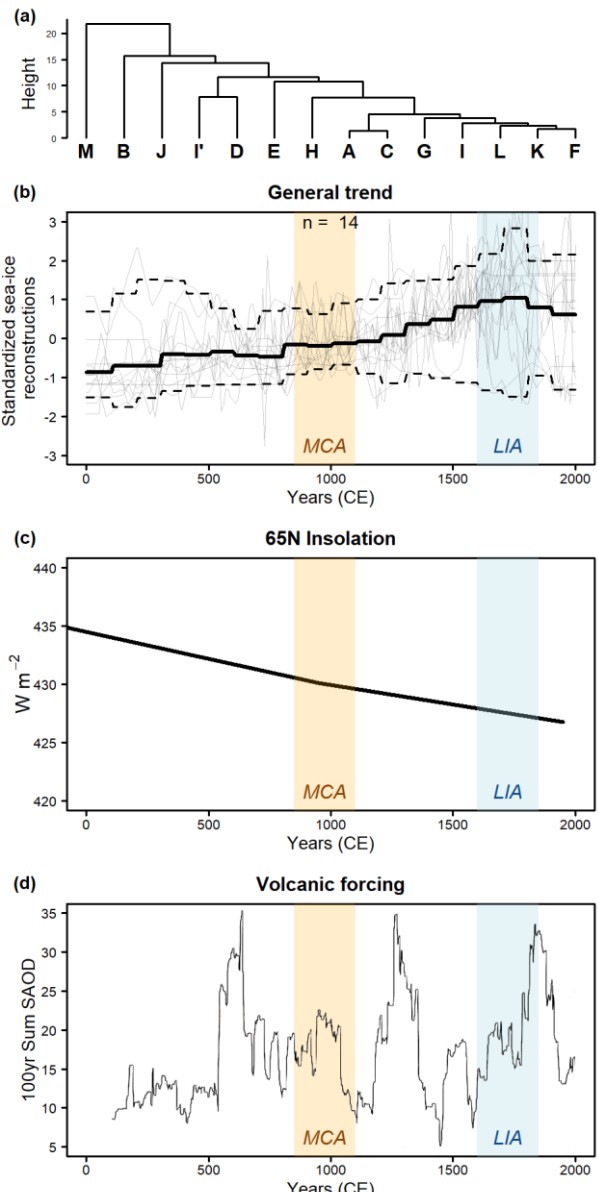

**Figure 2. (a)** Dendrogram from cluster analysis and **(b)** average composite of the standardized sea-ice reconstructions used in this study. The solid line represents the mean values of all records, the thin grey lines show the individual standardized sea-ice reconstructions, and the dashed lines represent the amplitude of the standardized values after binning to 100-year time slices. All records witness a common sea-ice expansion trend until around 1850 CE, suggesting that they were under the influence of the same dominant forcing. **(c)** Insolation at 65°N from Berger and Loutre (1991). **(d)** Total Stratospheric Aerosol Optical Depth – SAOD from Büntgen et al. (2020). Larger SAOD values are indicative of larger volcanic dust inputs to the atmosphere. The translucid orange and blue areas indicate the Medieval Climate Anomaly (MCA) and Little Ice Age (LIA) for the Arctic region (Wang et al., 2022).

In addition to the common trend, the individual proxy records presented a large variability, especially in the first and last five centuries of the Common Era (Figure 2b). The variability observed in the sea-ice reconstructions is probably caused by other environmental changes than the large-scale expansion/retraction of sea-ice cover. On both sides of Greenland, for example,

the intrusion of warm Atlantic water has been suggested as the cause of the observed short-term variability in proxy-based sea-ice records (Kolling et al., 2017; Jackson et al., 2021). Changes in the position of the atmospheric polar front (Allan et al., 2018) and interactions with ice produced within fjords (Kolling et al., 2017) have also been suggested as explanations of this variability.

Although all records witness a common forced trend, a large spread in records in the early and late part of the Common Era

indicate the presence of other local variability. To identify such second and higher order variations, we removed the common nonlinear trend (the average composite observed in Figure 2b) from all records and repeated the cluster analysis with the now regionally detrended data. When the general trend in the proxy records was removed, three distinct groups emerge from the cluster analysis (Figure 3a). Importantly, the three groups (types of sea-ice evolution) do not form geographical entities. Instead, the sites within each group have sites located on both the western and eastern side of Greenland.


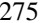

**Figure 3. Dendrogram (a) and average composite (c, d, e) of the standardized sea-ice reconstructions used in this study after removing the main trend (global average). The solid thick lines represent the mean values of all records, the thin lines show the individual standardized sea-ice reconstructions, and the dashed lines represent the amplitude of the standardized values after binning to 100-year time slices. The spatial distribution of records in each cluster is presented in (b). The solid thick black line in (b) represents the spatial domain used in the climate simulations. The translucid orange and blue areas in plots (c), (d) and (e) indicate the Medieval Climate Anomaly (MCA) and Little Ice Age (LIA) for the Arctic region (Wang et al., 2022). Most records suggest a sea-ice expansion (G1 and G2), followed by a retraction after the 18th (G1) and 16th (G2) centuries. G3 groups local records from scattered locations that do not follow the previously described sea-ice expansion trend in G1 or G2.**

The first group (G1; Figure 3c and Supp. Table S1) presented practically the same pattern as in the general trend (Figure 2), with maximum sea-ice cover around 1750 CE, corresponding with the LIA in the Arctic (1600 – 1850 CE; Wang et al., 2022). Group 1 includes three records from Baffin Bay, two records retrieved off Svalbard and two records from northern Iceland. While Baffin Bay is currently under the influence of seasonal sea-ice cover, the other four records were collected in regions south of the current winter sea-ice edge (Figure 1). The sea-ice expansion demonstrated by G1 until 1750, even after removing the general trend (Figure 2), highlights the importance of large-scale processes at these sites, such as transient changes in orbital forcing in combination with volcanism causing atmospheric cooling and consequent sea-ice expansion. Massé et al. (2008) reported the highest $IP_{25}$ concentrations over the past millennium during the 17th and 19th centuries in the Iceland Sea, in agreement with maximum sea-ice area at that time. Interestingly, G1 demonstrates a higher rate of sea-ice increase from 1600 CE until the end of the LIA, which is not apparent in the general trend (Figure 2). While the sea-ice proxy compilation suggests a rather gradual cooling trend, episodic snowline lowering and ice cap expansion in Arctic Canada highlight that part of the cooling trend is enhanced during periods of explosive volcanism (Miller et al., 2023) that is not clearly visible in nearby sea ice.

Considering individual records and local variability, Cabedo-Sanz et al. (2016) and Miles et al. (2020) suggest that this apparent accelerating expansion of sea ice south of the Fram Strait might be caused by more sea-ice export by stronger southward currents (mainly the East Greenland Current) carrying polar waters and pack ice. Thus, in addition to the sea-ice growth caused by the Neoglacial cooling, ice was also transported to more southern latitudes by stronger currents. Following the maximum sea-ice cover during the LIA, a slight decreasing trend in sea ice is demonstrated by G1. However, out of the seven records in this group, only three (A, G and L) cover the modern industrial period (after 1850 CE) (Supp. Figure S1). While record G shows a decrease in the sea ice (Cabedo-Sanz et al., 2016), the other two records point to highly fluctuating sea-ice conditions (Köseoğlu et al., 2018; Jackson et al., 2021). Thus, the plateauing trend in the last century might be an artifact of the analysis, since four of the records did not have an adequate time coverage for the modern period (Supp. Figure S1).

Group 2 (G2; Figure 3a) is a cluster of four proxy records collected in the Fram Strait (B, D), off eastern Greenland (E) and near Davis Strait, off western Greenland (I'; Figure 3b). This group shares the general trend of sea-ice expansion in the beginning of the Common Era with G1 (Figure 2), but the maximum values are reached a few centuries earlier, between 1250 and 1650 CE. In addition, the decreasing trend following the sea-ice maximum is more distinct (Figure 3d and Supp.

Table S1). A similar pattern was observed in polar and subpolar species of foraminifera from the Fram Strait (core MSM05/5_712-1; Spielhagen et al., 2011). The relatively similar long-term trend between G1 and G2 until ~1500 CE suggests a common response to a large-scale climate forcing where G2 includes temporal deviations from the overall similar trend. For the past 500 years, G2 diverges from the continued long-term trend in G1 showing an earlier decrease in sea-ice. Differing internal system responses and methodological constraints of $IP_{25}$ proxy may have an influence on this deviation. The early sea-ice maximum in G2 was followed by a sea-ice decrease, especially in the two records from Fram Strait (B and D). This decrease is probably an artifact caused by the two end-member scenarios for absent $IP_{25}$ in downcore records (Belt, 2018). While $IP_{25}$ (produced by sympagic diatoms) is absent in ice-free oceans, thick, perennial sea ice also reduces sympagic diatom and thus $IP_{25}$ production (Belt, 2018). Considering the record in their study, Cabedo-Sanz and Belt (2016) suggested that a switch from long seasonal sea-ice cover to a perennial sea-ice scenario could explain the early decrease in $IP_{25}$ observed in Fram Strait. Another explanation by Kolling et al. (2017) and Syring et al. (2020) also suggests colder climate but coupled to more open-water conditions due to polynya (ice free area) formation on the East Greenland shelf. It was probably caused by glacier expansion and katabatic winds pushing sea ice off the coast.

Considering the whole studied area in our study, the explanation of a cold climate but with ice-free areas also fit with the findings from the foraminifera data of Spielhagen et al. (2011), where the high fluxes of polar and subpolar species during the early LIA were associated with sea-ice margins. The apparent sea-ice retreat observed at I´ in Davis Strait, the only G2 site in the western side of Greenland, might be explained by the establishment of polynya conditions and an increase in sea-ice production along Greenland's western side. Polynyas are areas of open water and thin ice in regions otherwise covered by sea ice and are common around Greenland (Smith Jr and Barber, 2007). The decrease in *Islandinium minutum* and *Islandinium? cezare* is concomitant with an increase in the abundances of phototrophic taxa (especially *Pentapharsodinium dalei*), potentially indicating a shift towards longer open-water seasons (Allan et al., 2018).

Interestingly, some records from sediment cores retrieved from areas very close to each other were not necessarily assigned to the same cluster group (Figure 3b). For example, a sea-ice decrease over the last two centuries was not observed in record I, a sea-ice reconstruction from the same sediment core as dinocyst record I' (MSM05/3_343310), but based on $IP_{25}$ (Kolling et al., 2018). The differing cluster allocation of the dinocyst record (I'; G2) and the $IP_{25}$ record (I, G3) is caused by the difference in their temporal trends and temporal coverages. Although having a good temporal resolution of 14 years, record I ($IP_{25}$) only covers the period until 1809 whilst record I' (dinocysts) extends to 2006 (Supp. Figures S1 and S2). Therefore, record I did not have an appropriate temporal coverage to register the potentially same decrease as observed in record I'.

The lack of temporal coverage is also the probable reason behind the assignments of records B and C in different groups, even though both sediment cores were retrieved from the eastern Fram Strait off Svalbard (Figure 3). While record B was assigned to Group 2 due to the decrease in $IP_{25}$ after the 15[th] century (Supp. Figure S2), record C was placed in Group 1. However, record C only covers the time until 1770 CE (Supp. Figure S1). Therefore, it does not have enough temporal coverage to register a potential $IP_{25}$ decrease.

Lastly, Group 3 (G3; Figure 3a) is formed by three records and presents no clear long-term trend. This group contains records that did not fit into the two previous groups rather than records with a similar sea-ice evolution (Figure 3e and Supp. Table S1). Sediment core OD1507 (composite 03TC-41GC-03PC) (record M in Figure 3b) is located inside a fjord and presents a long-term decrease in $IP_{25}$ concentrations (Detlef et al., 2021). The authors of the original study argue that production of $IP_{25}$ decreases with the establishment of near perennial sea-ice cover during the Common Era. Therefore, we consider that the decrease in $IP_{25}$ is probably caused by the two end-member scenarios for absent $IP_{25}$, since perennial sea ice reduces sympagic blooms and IP25 production (Belt, 2018). Sediment core DA06-139G (record J in Figure 3b; Supp. Figure S3) was also collected inside a strait, and was thus under the influence of land-fast sea ice and marine-terminating glaciers (Andresen et al., 2011). The lack of temporal trend might be related to the influence of freshwater coming from the continental ice sheets where atmospheric warming may increase melting and adversely cool the surface waters locally as can be observed in glacier-fed lakes during heatwaves (Kirchner et al., 2021). Limoges et al. (2018) advise about the use of $IP_{25}$ in coastal areas, as lower salinities may influence the structural nature of sea ice and its propensity to sustain algal growth. Therefore, freshwater influence might cause a decline in $IP_{25}$-producing diatom species.

The sediment core MD99-2263 (record H in Figure 3b), on the other hand, is located off the Iceland shelf in an open water environment and close to cores JR51-GC35 (F) and MD99-2269 (G). Putting the record in context along with these nearby records in our compilation, the main difference between records F and G (both assigned to Group 1) and record H (assigned to group 3) occurred between 1450 and 1650 CE (Supp. Figures S1 and S3). During this period in record H, high sediment accumulation rates coincided with relatively high concentrations of $IP_{25}$ (Andrews et al., 2009). The authors suggest that in this location the sea-ice proxy $IP_{25}$ was associated with increasing drift ice related to changes in the wind pattern during the European LIA. Stronger N and NW winds led to more sea ice being exported from the Arctic Ocean towards the NW/N Iceland shelf. However, this increment was not as noticeable as observed in the records F and G. One possible explanation for this difference is the spatially diverse impact of stronger currents carrying warm waters on this region. Sediment core MD99-2263 (record H) was collected in the Denmark Strait, in a region directly under the influence of the North Irminger Current that carries warm Atlantic waters towards the Nordic Seas (Talley et al., 2011a). Sediment cores JR51-GC35 (F) and MD99-2269 (G), however, are located in a more protected region off northern Iceland. Between 1450 and 1650 CE, Europe was already under the influence of LIA unlike Greenland and North America (Wang et al., 2022). The difference in the temperatures caused a strengthening of the subpolar gyre, resulting in a stronger presence of warm and saline Atlantic water carried by the Irminger Current in the Denmark Strait region (Miettinen et al., 2012), where sediment core MD99-2263 (record H in Figure 3) was retrieved. Thus, it is possible that during this period, the presence of warm waters intensified the melting process of drift ice, promoting the transfer of sea ice proxies from the ocean surface to the sediments. Around 1600, the onset of LIA in Arctic and North America (Wang et al., 2022) probably caused a weakening of the Atlantic currents towards the Nordic seas.

## 3.2 Model-based sea-ice reconstructions

Annual mean air temperature trends (Figure 4) produced by both models in our study present a similar declining pattern as
proxy-based Arctic temperatures obtained in a reconstruction by McKay and Kaufman (2014). The McKay and Kaufman (2014) reconstruction obtained a cooling of -0.49 °C/kyr for the Arctic region (latitude > 60° N) for the pre-industrial part of the Common Era (Supp. Figure S4). A comparable but less pronounced cooling is simulated by the CESM1 and MPI-ESM models, with cooling rates of -0.28 and -0.21 °C/kyr, respectively (Supp. Figure S4). This implies that the reconstructed cooling trend is roughly twice as strong as in both simulations. After 1850, air temperatures present a shift towards a
warming trend. In the McKay and Kaufman (2014) proxy reconstruction, the warming is 0.97 °C/100 years, while the CESM1 and MPI-ESM models yield warming rates of 1.34 and 0.84 °C/100 years, respectively (Supp. Figure S4).

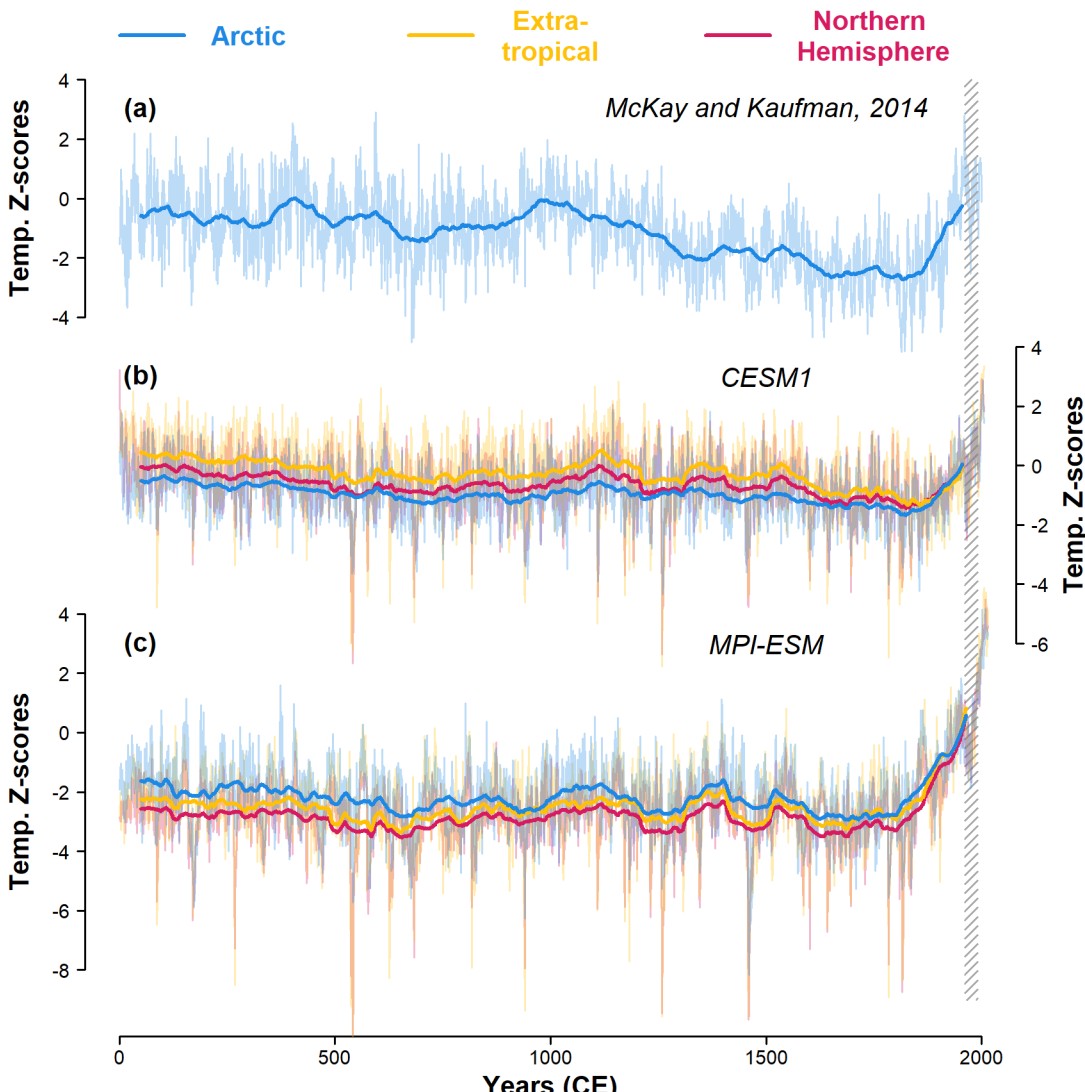

**Figure 4.** Air temperature *z*-scores (relative to the period between 1961-1990) based on (a) Arctic proxy data (McKay and Kaufman, 2014), (b) CESM1 simulation data and (c) MPI-ESM simulation data. Bold lines refer to 100-years running means. The hatched area indicates the period between 1961-1990. Northern Hemisphere refers to latitudes between $0 – 90°$ N, the Arctic region refers to latitudes between $60 – 90°$ N, and the Extra-tropical region refers to latitudes between $30 – 60°$ N. Temperature simulations from both models and for all three regions display a very similar long-term evolution, showing cooling temperatures for most of the Common Era, followed by a warming trend after 1850. This is consistent with the leading cluster G1 for the sea-ice proxies in Figure 3c.

Considering the whole Arctic region, sea-ice area from both models for the period 0 - 2000 CE, CESM 1 shows 27.4% more summer sea ice than MPI-ESM (average areas: 6.6 million $km^2$ vs 5.2 $km^2$, respectively). During winter, the difference is very small and decreases to 1.7%, with CESM1 presenting 16.1 million $km^2$ of average sea-ice area and MPI-ESM, 15.9

million km$^2$ (Table 2). There is less consistency in the region around Greenland with a strong difference during summer, with CESM 1 showing 73.4% more summer sea ice than MPI-ESM (average areas: 1.4 million km$^2$ vs 0.8 km$^2$, respectively) (Table 2). However, disregarding the magnitude and spatial bias of the sea-ice area, temporal changes in summer sea-ice area simulated around Greenland follow the changes as simulated for the whole Arctic Ocean (Figure 5). Considering the 100-year running mean, the correlation of annual sea-ice area between the total Arctic Ocean and the region around Greenland is

0.92 for CESM1 and 0.93 for MPI-ESM during summer, and 0.57 for CESM1 and 0.94 for MPI-ESM during winter.

**Table 2. Comparison between simulated sea-ice areas (in million km$^2$) between the two models (CESM1 and MPI-ESM) for the Northern Hemisphere and for around Greenland (see mask in Figure 1) for annual, summer and winter areas, over the Common Era (1 – 2000 CE).**

| Sea-ice area (10$^6$ km$^2$) during the Common Era | | Northern Hemisphere | | | Greenland | | |
|---|---|---|---|---|---|---|---|
| | | Minimum area | Annual average area | Maximum area | Minimum area | Annual average area | Maximum area |
| CESM1 | Summer | 4.7 | 6.6 | 8.3 | 0.9 | 1.4 | 2.2 |
| | Winter | 15.0 | 16.2 | 17.6 | 4.1 | 4.6 | 5.2 |
| MPI-ESM | Summer | 3.6 | 5.2 | 7.0 | 0.4 | 0.8 | 1.3 |
| | Winter | 13.8 | 15.9 | 18.2 | 3.5 | 4.2 | 5.5 |

When compared to the satellite data (Doerr et al., 2021), both models show slightly smaller summer sea-ice areas for the period between 1979 and 2000 (Supp. Table S2). While the numerical model simulated minimum summer sea-ice areas around 4.8 million km$^2$ (CESM 1) and 3.6 million km$^2$ (MPI-ESM), sea-ice area calculations based on satellite data were around 5.4 million km$^2$. The winter sea-ice areas, however, presented a good agreement between the results of the numerical

model simulations and the satellite data. The numerical model simulated maximum winter sea-ice areas around 16.2 million km$^2$ (CESM 1) and 15.1 million km$^2$ (MPI-ESM), and sea-ice area calculations based on satellite data were around 15.4 million km$^2$.

# Sea ice area ($10^6$ km$^2$)

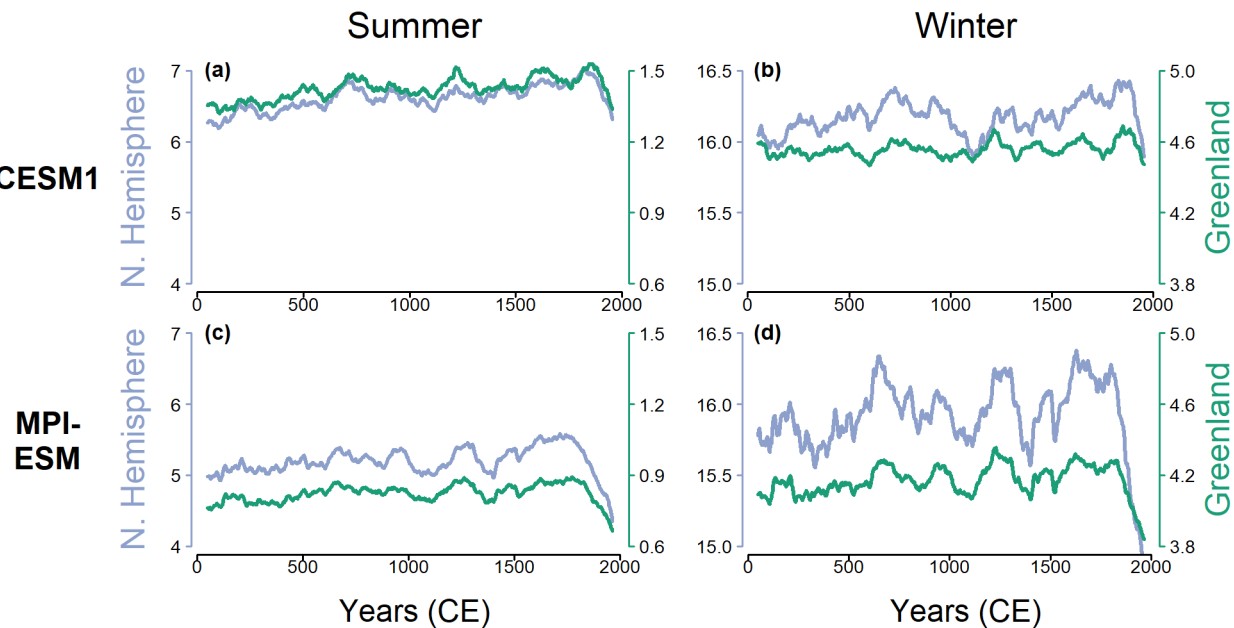

**Figure 5. Sea-ice area (in million km²; 100-year running mean) in both models, around Greenland (green) and in the whole Northern Hemisphere (blue), for boreal summer and winter seasons, over the Common Era. The very high agreement indicates that sea-ice proxy-compilations around Greenland are representative for the total northern hemispheric sea-ice area on longer time scales. Disregarding magnitude bias, temporal changes in the sea-ice area simulated around Greenland follow the changes as simulated for the whole Arctic Ocean, especially during summer.**

A general positive trend of increasing sea ice is simulated until 1850 for both seasons, followed by a sharp decrease in the sea-ice area due to global warming (Table 3). During the pre-industrial period of the Common Era, sea-ice areas simulated for the whole Northern Hemisphere suggest expansion rates around 0.02 million km²/100 years, and around 0.005 million km²/100 years when simulated for the region around Greenland. Nevertheless, the expansion rates are so small that when the maximum areas were compared over the centuries before 1850, the maximum sea-ice cover expanded less than 1% per 100 years during the Common Era (Supp. Figure S5). After 1850, however, the trend shifted and both models registered decreases in sea-ice area. In the whole Northern Hemisphere, the simulated sea-ice area shrunk around 0.7 million km²/100 years (Table 3), corresponding to a loss ~ 9% area/100 years during summer and ~ 4% area/100 years during winter considering the maximum sea-ice area during the Common Era. Around Greenland, simulations suggest sea-ice shrinkage of around 0.2 million km²/100 years after 1850 (Table 3). When compared to the maximum sea-ice area simulated for the Common Era, it represents a decrease of ~ 11% area /100 years in summer and ~ 6% of loss area /100 years in winter (Supp. Figure S5).

**Table 3. Comparison between trends of simulated sea-ice area (in million km²/100 years) between the two models (CESM1 and MPI-ESM), in Northern Hemisphere and around Greenland (see mask in Figure 1), and for boreal summer and winter seasons, over the Common Era (1 – 2000 CE). The % values consider the maximum sea-ice area simulated for the CE as reference.**

| Trend ($10^6$ km² / 100 years) | | Northern Hemisphere | | Greenland | |
|---|---|---|---|---|---|
| | | 1 – 1850 CE | 1851 – 2000 CE | 1 – 1850 CE | 1851 – 2000 CE |
| CESM1 | summer | 0.028 | -0.762 | 0.006 | -0.260 |
| | | (0.3%) | (-9.2%) | (0.3%) | (-11.9%) |
| | winter | 0.009 | -0.777 | 0.001 | -0.328 |
| | | (0.1%) | (-4.4%) | (0.0%) | (-6.3%) |
| MPI-ESM | summer | 0.021 | -0.627 | 0.005 | -0.138 |
| | | (0.3%) | (-8.9%) | (0.4%) | (-10.8%) |
| | winter | 0.021 | -0.692 | 0.008 | -0.278 |
| | | (0.1%) | (-3.8%) | (0.2%) | (-5.1%) |

To identify the main leading modes of sea-ice variability and their spatial loading patterns, we performed an EOF analysis for simulated sea-ice data after log-transforming the fractional data. Overall, the EOF analysis using both the 100-year running mean and the 100-years mean chunks data yield very similar spatial and temporal patterns in the first mode of variability (EOF 1 and PC1; Supp. Figures S6 and S7). This similarity shows that the approach of creating 100-years mean chunks for the proxy data will retrieve the main temporal behaviour and likely does not create artificial patterns in the EOF analysis. As the sea-ice proxies are mainly related to the period of sea-ice melt, we will focus the discussion on summer sea ice, i.e., minimum annual area. Figure 6 shows the first and second mode of variability of summer sea ice. While the principal components' plots (PC1 and PC2) present the results from both 100-year running mean and 100-years mean chunks data, the EOF1 and EOF2 plots only present the 100-year running mean results.

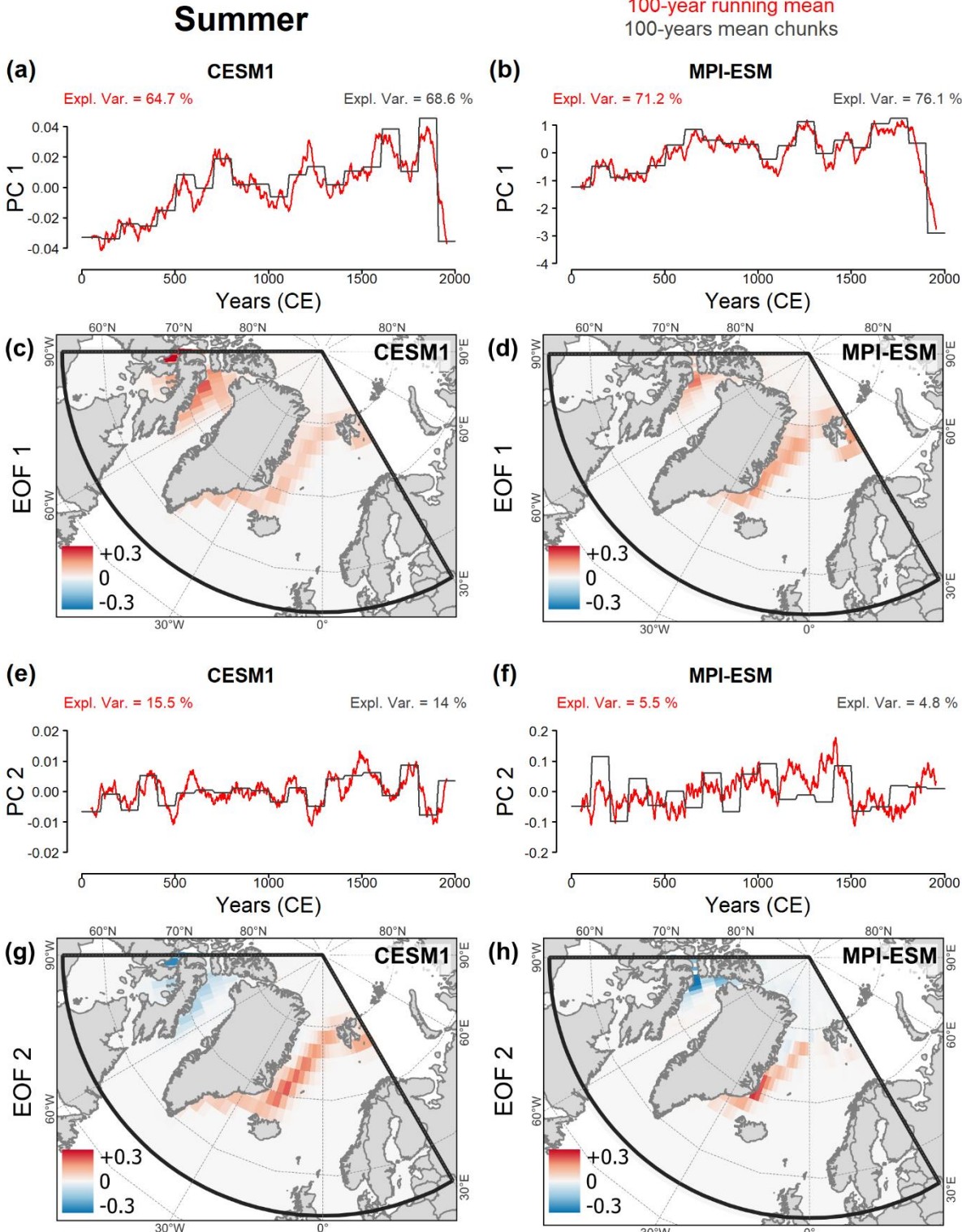

**Summer**

100-year running mean
100-years mean chunks

(a) CESM1
Expl. Var. = 64.7 %    Expl. Var. = 68.6 %

(b) MPI-ESM
Expl. Var. = 71.2 %    Expl. Var. = 76.1 %

(c) CESM1

(d) MPI-ESM

(e) CESM1
Expl. Var. = 15.5 %    Expl. Var. = 14 %

(f) MPI-ESM
Expl. Var. = 5.5 %    Expl. Var. = 4.8 %

(g) CESM1

(h) MPI-ESM

**Figure 6. Results of empirical orthogonal function (EOF) analysis of the log-transformed sea ice fraction based on CESM1 (a,c,e,g) and MPI-ESM (b,df,h) data. Normalized principal components plots considering the 100-year running mean data (in red) and 100-years mean chunks mimicking proxy data resolution (in dark grey); PC1: (a) and (b); PC2: (e) and (f). First and second EOF mode loadings plots consider the 100-year running mean data; EOF1: (c) and (d); EOF2: (g) and (h). In both models' simulations, the first mode captures most of the variability, with a sea-ice expansion trend during most of the Common Era followed by a sharp retraction in the last two centuries being simulated for the Labrador and Nordic seas. The second mode has a less clear temporal trend but displays an opposite pattern between the Labrador and the Nordic seas.**

PC1 from both models (CESM1 and MPI-ESM) show the expected long-term trend of increasing summer sea ice until around 1850 as in Figures 5a and 5c, followed by a sharp decrease (Figures 6a and b). The first leading mode explains more than 65% of the variability in the CESM1 model and more than 70% of variability in the MPI-ESM model. Spatially, most of the variance on multidecadal to centennial scale is concentrated in the Canadian channels and the Greenland Sea in both models (Figures 6c and d). PC2, on the other hand, displays a temporal pattern governed by internal variability and explains only around 15% and 5% of the variability in CESM1 and MPI-ESM models, respectively (Figures 6e and f). Although the temporal trends (PC2) were not quite similar between both models, the spatial structure of EOF2 presented the same opposite pattern between Baffin Bay and Greenland Sea in both models (Figures 6g and h). This internal or residual variability might be related to changes in the sea-ice export through the Fram Strait. Given the low explained variance in the simulations, we would not expect a large explanatory value in the noisier sea-ice proxy data. Thus, the first leading mode appears to contain most of the interpretable variance on larger than centennial scales over the last two millennia for the chosen area.

In addition to the increasing summer sea-ice trend prior to 1850, PC1 in both models contains some quasi-periodic variability. This was also present in PC2. A wavelet analysis of the annual summer sea-ice area identified the main periodicities (Figure 7). For CESM1, the multidecadal quasi-periodic variability is centred around 21, 33 years and 74 years, and for MPI-ESM around 27, 41 and 74 years.

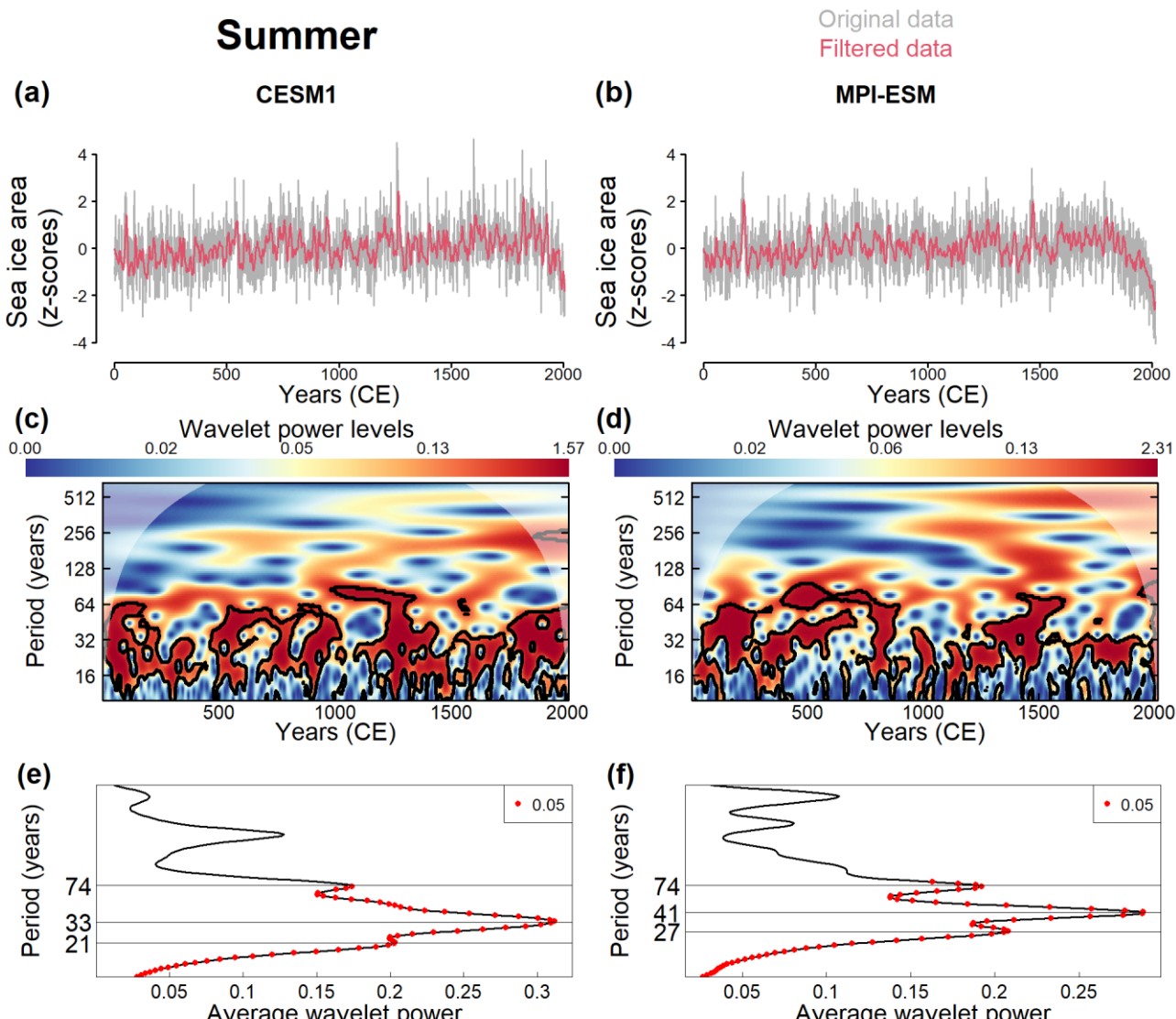

**Figure 7. Wavelet analysis from both numerical models (CESM1 and MPI-ESM), considering summer sea-ice area. (a,b) Normalized time series of sea-ice area considering the original data (in grey) and the filtered data (in red). (c,d) Wavelet power spectrum and (e,f) global wavelet of the normalized signal on the time-frequency domain. Marked regions on the wavelet spectrum indicate significant power to a 95% confidence interval. The areas under the cone of influence show where edge effects are important. Sea-ice simulations from both models display significant multidecadal quasi-periodic variability but no significant**
**(multi-)centennial variability as found in longer simulations and proxy records for the Holocene (e.g., Askjær et al., 2022).**

One possible explanation for this multidecadal variability could be related to solar cycles. Solar activity has been reconstructed from measured concentrations of [14]C in tree rings and [10]Be in the GRIP ice core (Usoskin et al., 2006; Vonmoos et al., 2006) and other sources that are also used in the forcing of past2k simulations used here (Jungclaus et al.,

2017). During the last 2000 years, one of the most pronounced periodicities is centred around ~80-year Gleissberg cycle

(Ma, 2009). During periods of high solar activity, summer sea ice might have presented smaller areas probably due to enhanced melting. However, another possible explanation for the multidecadal cycles might be related to the Atlantic Multidecadal Oscillation (AMO; Schlesinger and Ramankutty, 1994). The North Atlantic sea-surface temperature exhibits a multi-decadal variability that cannot be fully explained by the radiative forcing (Ting et al., 2014) and is probably related to the atmospheric-oceanic teleconnections affecting the heat transport in the North Atlantic (Divine and Dick, 2006; Fang et al., 2022). Positive phases of AMO have been linked to a greater penetration of warm subtropical waters into the subpolar gyre (Buckley and Marshall, 2016), causing sea-ice retreat especially in the Atlantic sector of Arctic Ocean (Miles et al., 2013; Kumar et al., 2021).

### 3.3 Long-term trend vs short-term fluctuations

One of the aims of this study was to assess how well individual sea-ice proxy records are suited to reconstruct long-term sea-ice trends and possibly multidecadal to centennial-scale variations that can be compared with climate model simulations. Based on the proxy-based reconstruction compilation, it is evident that when clustered together, the common trend in proxy records agrees well with the simulated summer sea-ice evolution. Both the reconstructions based on marine proxy data and on numerical models demonstrate a long-term increase in sea-ice cover until around 1850, followed by a decrease in the modern industrial period. This same behaviour was observed by Brennan and Hakim (2022). This increasing trend largely agrees with recent temperature reconstructions compilations (McKay and Kaufman, 2014; PAGES 2k Consortium et al., 2019), which observed a cooler period between 15th and 18th centuries followed by a sharp warming especially during the 20th century. However, there is discrepancy in the point of time when the long-term sea-ice trend changes. While in the model reconstructions the sea-ice retreat begins clearly only in the 20th century (Figures 5a, 5c, 6a and 6b), in the proxy records a sea-ice retreat is apparent immediately following the LIA (from the 18th century; Figures 2a and 3c). This retreat after 1850 was observed by Divine and Dick (2006) as a retreat in the sea-ice edge position in the Nordic Seas.

Interestingly, although individual proxy records presented some internal variability, they did not coincide with the multidecadal-scale fluctuations observed in the numerical model reconstructions. In the records that presented significant ($p < 0.05$) average wavelet power, there was no consistent variability among the proxy records (Supp. Figures S8-S14). This difference is likely caused by internal climate and ecosystem dynamics where proxy-based sea-ice reconstructions were affected by complex environmental parameters, while the physical models directly responded to large-scale atmospheric forcing from variations in solar, volcanic, and orbital forcing. Furthermore, while the proxy records used here have relatively high resolution (< 100 years), the resolution is generally not enough to capture multidecadal oscillations. Additionally, uncertainties related to the core sub-sampling and the age-depth models can cause chronology errors that might affect the wavelet results. Lastly, caution should be exercised when interpreting these values, since they can be an artifact of the wavelet analysis itself. Most of the observed significant periods coincide with portions of the sediment cores that have relatively high time resolution. Therefore, what was identified as significant periods by the wavelet analysis might have been caused by changes in sampling resolution rather than changes in natural signals. This implies that the absence of significant

multi-decadal variability in local proxy records is rather caused by low sampling resolution while periods with high resolution confirm the dominance of multidecadal variability (Supp. Figures S10 and S12), as captured in both model simulations (Figure 7; note that model data was filtered to remove the sub-decadal variations prior to the wavelet analysis).

In addition to forced variability creating some co-variability between both simulations on multi-decadal to centennial timescales (Figure 6), another explanation might be related to internal variability of the numerical models. The spatial structure of EOF2, associated with the multidecadal to centennial-scale fluctuations, in both models presented a similar dipole pattern, with positive values observed in the Greenland Sea and negative values observed in the inner Baffin Bay. The similarity in their spatial structure might be a methodological effect of the orthogonality required by the EOF analysis. While the first EOF presents a unipolar spatial pattern, the second EOF will present a bipolar spatial pattern; in this case, to the east and west of Greenland, with loading patterns with opposite signs. However, their temporal patterns were not alike (Figures 6e and 6f). This inconsistency is probably caused by the low explained variance of PC2 (15% for CESM1 and 5% for MPI-ESM), incorporating a large degree of residual noise. Although the multidecadal and centennial variability of Arctic summer sea ice has been linked to changes in the northward Atlantic and Pacific heat transport and in the Arctic dipole pattern, there is still some significant variability between mean states of Arctic sea ice simulated by different models (Li et al., 2018).

Another explanation for the discrepancy observed in the multidecadal to centennial-scale fluctuations between proxy-based and numerical model reconstructions might be related to seasonality. $IP_{25}$ is generally produced during the sympagic spring bloom prior to ice melt (Belt, 2019), while *I. minutum* cyst production probably occurs in the water column over the open water season (Heikkilä et al., 2016; Luostarinen et al., 2023). Thus, sea-ice proxies do not register sea ice in similar manner, although all proxy-based sea-ice reconstructions resembled more the model-based summer season (annual minimum sea-ice area) than the winter season (annual maximum sea-ice area) simulations. Differences in the seasonal signal of sea-ice proxies can also cause inconsistent signals between palaeorecords (Kolling et al., 2018). Thick snow cover during the early part of the melting season can decrease sympagic algal (and hence $IP_{25}$) production due to light limitation (Leu et al., 2015; Lim et al., 2022). However, the same thick snow cover during the late melting season can also extend the period of dominance of ice algal blooms over that of phytoplankton (Leu et al., 2015). The cold freshwater layer from sea-ice melt in late spring and early summer, when light is not a limiting factor, might enhance the role of water-column diatom (*Fragilariopsis* spp., *Fossulaphycus arcticus*) and dinoflagellate (*Islandinium minutum*, *I.? cezare*) species used as sea-ice proxies.

Polynya formation can also add to the internal variability observed in the proxy-based reconstructions. In model simulations, however, much of the knowledge of the air–sea interaction over polynyas comes from atmospheric models, with not enough horizontal resolutions (often > 75 km) to properly resolve polynya dynamics (Moore and Våge, 2018). In turn, polynyas are often characterized as biological hot spots (Daase et al., 2020), with conspicuous sea-ice proxy production (Limoges et al., 2020; Harning et al., 2023).

Lastly, marine-terminating glaciers play an important role in coastal areas, creating possibly markedly differing sea-ice conditions and productivity when compared to sites further offshore (Kolling et al., 2018; Detlef et al., 2021; Jackson et al., 2021), adding to the internal variability of some proxy-based reconstructions. In these areas, glacier meltwater might

generate a cold and brackish surface-water layer, causing salinity-driven stratification, an environment preferred by some sea-ice proxies, such as the diatoms *F. oceanica* and *F. reginae-jahniae* (Weckström et al., 2020).

Overall, the cluster analysis across a compilation of proxies presented here shows high potential to further analyse the presence of multi-decadal to multi-centennial variations in addition to long-term trends on longer timescales than the last 2000 years, depending on the core resolution. Apart from some potential local dependencies in proxy data described here, we consider it highly likely that significant (multi-)centennial variability could be detected in longer proxy records which might improve the signal-to-noise-ratio in data. A recent assessment of over 120 records and several transient climate simulations

has revealed a clear global presence of significant (multi-)centennial oscillation with 100–200 years periods during the Holocene with the strongest power in Arctic regions (Askjær et al., 2022).

## 4 Conclusions

In this study, our goal was to explore the long-term trends and low-frequency variability in proxy-based reconstructions and to compare them with transient climate simulations. Both the clustered proxy-based sea-ice reconstructions and the model

simulations revealed a long-term increase in sea-ice cover over most of the Common Era. The maximum sea-ice cover was recorded around the 17$^{th}$ and 18$^{th}$ centuries, identified as the Little Ice Age (LIA), probably related to intense volcanic activity. The long-term sea-ice expansion over the Common Era has been largely explained by the decrease in the Northern Hemisphere insolation due to orbital changes. The recent sea-ice retreat, on the other hand, was likely initiated by a recovery after the LIA further amplified by anthropogenic greenhouse gas emissions towards the 20$^{th}$ century. Short-term variability,

however, was less coherent, whether among the proxy-based reconstructions, among the two numerical models, or between the model simulations and the proxy reconstructions. Therefore, local-to-regional scale forcings, internal variability and noise in the proxy data, coupled to poor temporal resolution as well as dating uncertainties, have a marked impact on the interpretation of short-them variability.

The good correspondence between clustered proxy-based records and numerical model reconstructions suggests that the

state-of-the-art sea-ice proxies are able to capture large-scale climate forcings consistent with externally forced climate models. However, some of the individual sea-ice proxy records incorporate local-scale environmental forcings that partially override large-scale forcings. Thus, considering the ecology of the proxy matters: different proxy types represent different habitats and seasons, and incorporate noise from environmental parameters other than sea ice, especially at coastal locations. Therefore, studying sea-ice history of a particular region should ideally be based on several sea-ice proxy-based

reconstructions. We suggest making use of stacked sea-ice proxy records in future large-scale sea-ice studies, as they appear to show high agreement with transient climate model simulations on longer timescales. Local reconstructions are equally important for reconstruction of local-to-regional scale patterns that are not captured by stacked records or climate models.

**Code and data availability**

Climate Data Operator software was used for analysing model data with CDO version 1.9.10 (Schulzweida, 2021; https://code.mpimet.mpg.de/projects/cdo). The PMIP4 past2k simulations with MPI-ESM and CESM1 contribute to CMIP6 and can be retrieved via the Earth System Grid Federation network ESGF (e.g., https://esgf-data.dkrz.de/search/cmip6-dkrz/ or https://esgf.llnl.gov/, last access: 25 March 2022). The R environment was used to analyse the data using the packages dplyr (https://cran.r-project.org/web/packages/dplyr/index.html), Hmisc (https://cran.r-project.org/web/packages/Hmisc/index.html), ncdf4 (https://cran.r-project.org/web/packages/ncdf4/index.html), signal (https://cran.r-project.org/web/packages/signal/index.html), TSclust (https://cran.r-project.org/web/packages/TSclust/index.html), WaveletComp (https://cran.r-project.org/web/packages/WaveletComp/index.html), and zoo (https://cran.r-project.org/web/packages/zoo/index.html).

**Author contribution**

Conceptualization: M.H., A.D. and F.S.; Funding acquisition: M.H., F.S.; Methodology and data analysis, A.D., F.S. and K.P.; Writing – original draft preparation, A.D.; Writing – review and editing: A.D., K.P., F.S. and M.H.; Visualization: A.D; Project administration: M.H. All authors have read and agreed to the published version of the manuscript. A.D. is the corresponding author.

**Acknowledgements**

This study was funded by the Research Council of Finland (grants 296895, 328540 and 334509 to M.H.). F.S. received funding from the Swedish Research Council for Sustainable Development (FORMAS 2020-01000). The Swedish-Finnish collaboration was funded by the Bolin Centre for Climate Research at Stockholm University and Arctic Avenue - a spearhead research project between the University of Helsinki and Stockholm University. The analysis of MPI-ESM and CESM1 simulations were enabled by resources provided by the National Academic Infrastructure for Supercomputing in Sweden (NAISS) at the National Supercomputer Centre (NSC), partially funded by the Swedish Research Council through grant agreement no. 2022-06725. The model simulation with CESM1 was conducted with high-performance computing support from Yellowstone (ark:/85065/d7wd3xhc) provided by NCAR's Computational and Information Systems Laboratory, sponsored by the National Science Foundation.

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
