# Peer review of "Sea ice variations and trends during the Common Era in the Atlantic sector of the Arctic Ocean"

_EGUsphere, 2023_

## Author Response (AR1)

**RC1: 'Comment on egusphere-2023-1327', Stefan Kern, 09 Aug 2023**

Review of "Sea ice variations and trends during the Common Era in the Atlantic sector of the Arctic Ocean" by Lindroth Dauner, A. L., et al.

**Summary:**

This manuscript deals with the compilation, discussion and analysis of a set of marine sea ice proxy records investigating the approximate development of the sea ice cover in the Atlantic Sector of the Arctic Ocean during the past 2000 years - aka the common era. The existing records are analysed by statistical means such as cluster, EOF, and wavelet analysis to find out dominating modes of sea ice variability. The results are compared with climate simulations by two climate models of the CMIP6 group. Analysis and model results are in good agreement. The sea ice cover information inferred from the marine proxy records tends to reflect the general trends well. However, deficiencies are discovered when looking into shorter-term and/or regional fluctuations in sea ice cover as suggested by the marine proxy records. Potential causes of these deficiencies are discussed.

**General comments:**

I have two general comments.

One is that I found it difficult to figure out what are the contributions and findings of the author team and what is the information that was taken from other sources / previous work. Don't get me wrong here: You cite and give credits to this other work very well but this information is so much mixed with the description of your results that I found it hard sometimes to disentangle these.

> **Reply**: Thank you for the constructive review. We tried to clarify the results and discussion section to separate better the results from the original datasets and our compilations based on clusters (lines 2446-247, 266, 274, 304, 327, 332, 340-341, 355, 357 and 366-367 in the marked up version). See more detailed responses to your comments below.

The second general comment exactly hooks up on this last impression. I was wondering whether the overall results of your manuscript could be worked out more clearly by finding a way to better separate your results and their description / interpretation by you on the one hand from the critical discussion of the results and their limitations plus hypotheses developed based on combining your results with those of previous work on the other hand.

> **Reply**: See previous reply. We think it is important to discuss the original data and methodological discrepancies (see also comments from Reviewer 2), and thus we discuss the original data when interpreting sea-ice trends in the clusters produced here. Presenting the results from the local data, without directly interpreting it regarding its local informational value, would not be very meaningful.

**Specific comments:**

L67/68++: I can agree that there is truly very limited information about the past sea ice cover in the Arctic; there are however some useful extensions of information about the sea ice cover from the satellite era back into the past - especially in the European Arctic - based on observations. For instance: Divine and Dick, 2006, Historical variability of sea ice edge position in the Nordic Seas, J. Geophys. Res.-Oceans, 111; England et al., 2008, A millennial-scale record of Arctic Ocean sea ice variability and the demise of the Ellesmere Island ice shelves, Geophys. Res. Lett., 35; I am sure there are more.

There are also reconstructions of, e.g. the Arctic sea ice volume, based on numerical modeling carefully tuned to present day conditions (see Schweiger et al., 2019, Arctic sea-ice volume variability over 1901-2010: A model-based reconstruction, J. Climate, 32).

Not sure whether you'd consider it worthwhile to harvest archives for more published work of others in this respect. It might be quite interesting.

> **Reply**: The lines you mention list the very few *compilations of proxy-based sea-ice reconstructions*. It is true historical observations exist and we will mention this in the Introduction. However, because the focus of this study was on sea ice reconstructions for the whole Common Era, we did not include any data that didn't cover at least 80% of it. Historical reconstructions don't fit this criterion and therefore were not included in the statistical analysis. Unfortunately, the ice-shelve reconstruction focused more on the ice-shelve establishment than on sea-ice variability, and on longer time-scale trends. The Arctic sea-ice volume reconstructions from Schweiger et al (2019), on the other hand, dealt with sea-ice changes over the last century, over an annual to decadal timescale. Therefore, the timescales were not compatible with the criteria set (see methods). However, we included the study of the historical sea-ice edge position in the Nordic Seas in the discussion (lines 517-518 in the marked up version).

L184-186: "Sea-ice extents ... gives the annual extent". In the sea ice community, sea ice extent is defined as the sum of the area of all grid cells covered by at least 15% sea ice. In contrast, sea ice area is the sum of the area of all grid cells covered by any sea ice weighed with the actual sea ice area fraction. From what you describe you seem to have computed the sea ice area and you should name it like this henceforth - aka use "sea ice area" instead of "sea ice extent". You might need to correct this in the entire manuscript.

> **Reply**: Indeed, we used the sum of the area of all grid cells covered by any sea ice weighted with the actual sea ice area fraction. Thus, the terminology was corrected to "sea ice area" throughout the manuscript. Thank you for pointing it out.

Figure 1: Please note the source of the bathymetry used and also note where the arrows denoting the currents stem from. If you plotted the latter by yourself you might want to at least state which source you used as blueprint. What is the source of the sea-ice median extent?

**Reply**: All the sources were included in the revised figure caption.

Figure 2:

    - I was wondering whether it would make sense to explain in a bit more detail what a dendrogram in general and in this case tells us.

    **Reply**: We changed "Dendrogram" to "Dendrogram from cluster analysis" in the caption. And a more detailed explanation was added in the items "2.3. Statistical analyses" (lines 181-183 in the marked up version) and "3.1. Proxy-based sea-ice reconstructions" (lines 246-247 in the marked up version).

    - I can see that panel b) shows a standardized quantity that somehow seems to be related to sea ice (area?) reconstructions. But the y-axis annotation says "Standard deviation" and the text in the caption speaks of "average composite ... solid ... mean values of all records .... dashed values represent the amplitude". These descriptions seem not to match well with each other and some explanation might be helpful.

    **Reply**: The y-axis label was fixed to "standardized sea-ice reconstructions" and the individual reconstructions were plotted in the figure to improve the understanding. The same was done for Figure 3.

L225: "Thus, an increase ..." --> I agree that this interpretation can be made. I was wondering however, to what degree a change in the dominant ice type, i.e. from multiyear ice to seasonal ice, could also have resulted in an increase in the seasonal sea ice - being interpreted as an overall increase in sea ice - which would be contrary to what we have been observing over the past decades in the Arctic.

    **Reply**: Most of the cores are located in regions that have not been covered, during the Common Era, by multiyear sea ice (line 228 in the marked up version). The only cores located in areas that might have been influenced by multi-year sea ice are cores D and M. And, for these two cores, the change in the dominant ice type is included in the discussion, which was especially relevant for site M (composite core 03TC-41GC-03PC) (lines 355-358 in the marked up version).

L282: "The similarity ..." I am not so sure I go with this similarity here because G1 seems to have an accelerated increase from -1 to +2 until 1700-1800 CE and a rather moderate decrease afterwards while G2 kind of ramps up from -1 to near 0 around 300 CE, followed by a rather linear and comparably weak increase to values around +1 in1500 CE followed by a quite remarkable decrease to values below 0 until present day. The time of the maximum value appears to coincide with a remarkable ramp up in G1 values. This is what I see there and I am not sure this could be explained by large-scale climate forcing that easily.

    **Reply**: We agree that the sentence is a bit vague, and we now described the details more clearly in the revised version (lines 315-322 in the marked up version). There is a common long-term trend from around -1 to +1 until around 1500 (so ¾ of the Common Era) in both

clusters. G2 is less smooth compared to G1 including your mentioned jump at 300 CE and hence includes temporal deviations from the overall similar trend. A clear divergence dominates the last 500 years. We will edit the text accordingly. Partly the reason can be methodological, i.e., the sea-ice maximum was not captured by the IP$_{25}$ proxy.

L289: "It was probably caused by ... of the coast." --> I take this sentence as the example to express the impression that a lot of what is written in this paragraph is less the presentation of results but rather a lot of discussion and hypothetical statements.

> **Reply**: We intentionally chose to merge results and discussion. Representing only the results for such local variations would be very descriptive and we would need to repeat all details in the discussion again, if separated. In our view, these local variations are only interesting or relevant if they are directly explained. Hence the direct inclusion of original references and/or own explanations. We tried, however, to clarify the reasoning behind the "hypothetical statements" throughout the manuscript.

I note that section 3 is indeed entitled "Results and Discussion" but I was wondering whether a more distinct separation of what are your results (i.e. what is new) and what are the points of discussion and hypotheses where you mix in a lot of information from other authors. I find it difficult to focus and lose traction on the results of this paper.

> **Reply**: See previous reply. Our result is the synthesis of local data to identify common vs. local changes which still requires to account for the local information and/or specifics of the sea-ice proxy from original studies. We are wording out "the original authors" when we look at their interpretations of the data; we think it is important to carefully consider the data origins beyond the cluster behaviour. But we changed the wording in some places to highlight better our results/interpretations and the interpretations by the original authors.

If you would try to separate discussion issues more clearly from the results issues it might also become more clear what the different influencing factors as well as the various limitations of the approach used actually are. Among these would be the first-year ice - multiyear ice issue repeatedly mentioned, or issues like changes in the location of the land/ice - sea ice transition zone and thereby in the strength of katabatic winds / polynya existence by changes in ice sheet extent.

> **Reply**: See our response above.

Note: This comment applies also to the previous and the following paragraphs

Table 2: I was wondering whether you at all thought about comparing the output of these two models also against observations of the sea-ice concentration from satellites. There you could only use data from the late 1970ies onwards but it might provide you with an idea whether any of the models is potentially biased in its representation of the sea ice extent (or area?) See my previous comment about your description of how you compute sea ice extent as well.

Reply: We are not aware of any notable link between biases and differences in long-term variations and trends. However, we agree that it is a relevant point to add a verification of the simulations. We hence did such a comparison now (lines 418-424 in the marked up version) and added the comparison between the models' results and the satellite data, for the period between 1979 and 2000. We included the table as Supplementary Material.

| Sea-ice area ($10^6$ km²) between 1979-2000 | | Northern Hemisphere | | | Greenland | | |
|---|---|---|---|---|---|---|---|
| | | Minimum area | Annual average area | Maximum area | Minimum area | Annual average area | Maximum area |
| CESM1 | Summer | 4.8 | 6.0 | 6.6 | 0.9 | 1.2 | 1.5 |
| | Winter | 15.0 | 15.6 | 16.2 | 4.1 | 4.3 | 4.5 |
| MPI-ESM | Summer | 3.6 | 4.2 | 4.8 | 0.4 | 0.6 | 0.8 |
| | Winter | 14.1 | 14.6 | 15.1 | 3.5 | 3.8 | 4.0 |
| Satellite data | Summer | 6.2 | 7.1 | 8.0 | 1.7 | 2.3 | 2.6 |
| | Winter | 15.7 | 16.2 | 16.9 | 6.7 | 7.3 | 7.7 |

Table S2. Comparison between simulated sea-ice areas (in million km²) between the two models (CESM1 and MPI-ESM) and sea-ice extent (in million km²) from satellite data (obtained from the National Snow & Ice Data Center website) for the Northern Hemisphere and for around Greenland (see mask in Figure 1) for annual, summer and winter areas, over the period between 1979 – 2000 CE.

**Sea ice area ($10^6$ km²)**

[Figure]

Figure (extra). Sea-ice area (in million km²) in both models and from satellite data, around Greenland (green) and in the whole Northern Hemisphere (blue), for boreal summer and winter seasons, over the Common Era.

**Typos / editorial comments:**

Line 47: "preventing heat and moisture transfers" --> perhaps better: "reducing or even almost preventing heat and moisture transfers"

> **Reply**: The text was changed as suggested (lines 47-48 in the marked up version).

Line 95: Suggest to replace "high-resolution" by the actual resolution in years.

> **Reply**: We cannot inform one specific resolution value because each of the 14 sea-ice reconstructions have their own resolution, which also varied through the records. But the average resolution was added to Figures S8 to S14 in the Supplementary Material.

L104 / L141: "original authors." --> Not clear what this means. The co-authors of this manuscript? Are the data associated with this part of the archived data you downloaded or is this additional data. If the latter I recommend that you emphasize this more.

> **Reply**: In the first case (L104), it was additional data that were not found in the searched databanks (now line 106 in the marked up version). In the second case (L141), the original age-depth models were used (now line 145 in the marked up version). In both cases, the sentences were rephrased to improve readability.

L195/196: "offset of 0.5 to avoid zeroes" --> So you are actually working with sea ice concentration data sets that have values ranging between 0.5 and 1.5, is this correct?

> **Reply**: The offset is an arbitrary number to allow for a log-transformation of fractional data for statistical analysis (lines 207-208 in the marked up version). So, the used numbers are the log-values derived from the fractional data across the range 0.5 to 1.5. It is a common procedure when using fractional data like sea ice or similar.

L251/252: "common non-linear trend" --> Two questions here: 1) why non-linear? and 2) would you mind to share the values of these trends?

> **Reply**: The "non-linear trend" is the average composite observed in Figure 2 (line 276 in the marked up version). The values of all the composites were included in the Supplementary Material of the revised manuscript.

L277/278: "since most of the results ... period" --> You could put more emphasis on this issue by providing information and/or referring to the actual time coverage.

> **Reply**: We tried to emphasize the information regarding the cores that do not cover the last two centuries of the Common Era (lines 312-313 in the marked up version).

Figure 5: I strongly recommend to have identical ranges of the extent displayed at the respective axis, i.e. N. Hemisphere Summer sea ice extent has the same axis range, N. Hemisphere, Winter sea ice extent has the same axis range.

> **Reply**: The idea behind the variable Y-axis ranges was to emphasize the similarity between sea-ice evolution from the whole Northern Hemisphere and from the area around Greenland. However, we agree and set the same Y-axis ranges as suggested as it is still possible to observe the similarity between the two regions.

**CC1: 'Comment on egusphere-2023-1327', Kirsten Fahl, 03 Sep 2023**

Review of "Sea ice variations and trends during the Common Era in the Atlantic sector of the Arctic Ocean" (Lindroth Dauner, A. L., et al.)

As an organic geochemist, my comments will mainly relate to this field of the manuscript.

**Summary:**

The article of Lindroth Dauner et al. focused on the currently relevant issue sea-ice variation in the Northern Hemisphere during the Common Era (past 2000 yrs). The sea-ice variation studies were conducted using proxy-based sea-ice datasets (already published data) and by different statistical means. In this context, long-term trends and low-frequency variability show different results. While long-term trends of both approaches are in good agreement, the short-term variability results of both approaches are less coherent.

**Comments:**

- Generally, the manuscript is well written.

-The literature referenced is current.

- The database of proxy-based sea-ice reconstructions is solid and the generation of the data is consistent with established methods of organic-geochemical analysis.

> **Reply**: Thank you for the constructive review.

-The authors have used nearly all relevant proxy-based sea-ice reconstruction datasets currently available for the Common Era in this area. The only data set, I'm missing, is from Core MD99-2275 (Iceland) published by Massé in 2008 (Guillaume Massé, Steven J. Rowland, Marie-Alexandrine Sicre, Jeremy Jacob, Eystein Jansen, Simon T. Belt; Abrupt climate changes for Iceland during the last millennium: Evidence from high resolution sea ice reconstructions. Earth and Planetary Science Letters 269, 564-568).

> **Reply**: We looked at the Core MD99-2275 data. Unfortunately, it does not fit our time coverage requirements (80% of the Common Era), since the data ranges between 800 and 1950 CE. Adding shorter timeseries would deteriorate the consistency of long-term trends as these would then partly depend on data availability rather than only climate. But we now reference this publication in the discussion (lines 298-299 in the marked up version).

- Personally, I would have liked to see some additional explanations of the figures so that also non-specialists could somewhat better evaluate the inferences made based on the results. Thus, at its

present stage, it limits the readership, even though the topic is certainly of interest to a broader community due to its topicality.

> **Reply**: See our responses to Reviewer 2. We changed axis title names and add explanations to the captions.

- In the text, the authors mention the Little Ice Age several times. It would certainly be helpful if such events would be highlighted in the figures (e.g., fig. 2d). As an example, see Kolling et al. (2017) figs. 2 and 6. The same applies to any recognizable warming events such as the Medieval Climate Anomaly (especially since Wang et al. 2022 is cited).

> **Reply**: We added the indication of the Little Ice Age and the Medieval Climate Anomaly in Figures 2 and 3.

- The authors have critically discussed the differences between the matches (or non-matches) of the results of the proxy-based datasets and the modeling results of the long-term trends and short-term variability and substantiated them with recent publications.

> **Reply**: Thank you for your kind comment. We tried our best.

-The authors have also not failed to point out the problems of interpreting different proxies, which are, for example, due to different habitats and different seasons of reproduction of the producing/synthesizing organisms and are additionally influenced by sea-ice independent parameters. In this context, I would additionally recommend Spielhagen et al. 2011 (MSM5/5-712; Enhanced Modern Heat Transfer to the Arctic by Warm Atlantic Water, 28 January 2011, vol. 331, Science) for the discussion, even though foraminifera are not used as a proxy in this manuscript.

> **Reply**: We added the suggested reference in our discussion (lines 318-319, and 332-334 in the marked up version).

- From the proxy point of view, it would be desirable if at least one proxy record as an example of each of the three groups would be transferred from the supplement to the main body of the manuscript.

> **Reply**: To avoid bias when choosing the only one proxy record per group, we added all the standardized proxy records in Figure 3. For the individual records with their identification, the readers are then referred to the Supplementary Material.

One last comment to Chapter 2.1. (line 131-133): ........"When available"...... In case of "not available", does this mean, that you have used different units in some cases! This is not the usual procedure.

> **Reply**: In those cases where the data was not normalized by TOC, we used the concentration data ($\mu$g/g dw or ng/g dw). We changed the sentence to make the information clearer (lines 135-136 in the marked up version).

**RC2: 'Comment on egusphere-2023-1327', Dmitry Divine, 25 Sep 2023**

Review of "Sea ice variations and trends during the Common Era in the Atlantic sector of the Arctic Ocean" (Lindroth Dauner, A. L., et al.)

**Summary:**

The manuscript presents a compilation of marine sea ice proxies in order to study the development of sea ice conditions in the northern NA sector over the Common era. Common variability present in the proxy series is studied using clustering. Time variability on various timescales is further analyzed using wavelets. Results inferred from the analysis of proxy data are compared against PC of transient climate and sea ice simulations from MPI-ESM and CESM1 fully coupled models. Both proxies and models share common multicentenntial to millennial scale tendencies, but diverge on shorter timescales. Causal factors for the these discrepancies are discussed.

In addition to the valuable points already highlighted by two other reviewers, I would like to suggest the authors to pay attention to a few more comments indicated below:

> **Reply**: Thank you for the constructive review and additional comments.

**Comments:**

Line 60: "Most human observations of sea ice are derived from satellites and date back only to the 1970s"

The authors are advised to check the older publications of Vinje et al., 2001 (J Climate , V14, p 255-), and  Divine&Dick2006 (doi:10.1029/2004JC002851) where sea ice retreat after 1850 based on historical sea ice observations in the Nordic seas is discussed.

> **Reply**: We now mention historical sea ice observations and the references in the Introduction (lines 59-60 in the marked up version), and also in the Discussion (Section 3.3) (lines 517-518 in the marked up version).

Line 85: "…an integration of existing sea-ice reconstructions in a systematic way…"  consider adding "for the entire northern North Atlantic", since one can find earlier publications where regional scale compilations of sea ice proxies were made.

> **Reply**: The sentence was completed as suggested (line 86 in the marked up version).

Line 91: "… large model biases exist in Arctic regions, attributable to complex feedback framework…" consider adding "… and remaining deficiencies in the implementation of sea ice in GCMs… "

> **Reply**: The sentence was completed as suggested (lines 92-93 in the marked up version). We now also included a verification of the two used models vs. observations as suggested by the other reviewer.

Line 108: F. cylindrus was demonstrated to be rather a cold water species than the one associated with sea ice (see Oksman et al., 2019, von Quillfeldt (2004)). You mention it later in the text, anyways, so what was the point to list it here as one of the indicator species?.

> **Reply**: This is a good point, in some locations *F. cylindrus* could be a relevant sea-ice proxy and that is why we added it in the initial data search. But since we did not find these data, we edited the text as suggested (lines 111, and 124-127 in the marked up version).

Line 164: "These represent the percentage of ice cover in each grid square and both variables are on atmospheric grids"

"percentage of ice cover in each grid square" is associated with sea ice area.

Why atmospheric grid (lower 2 degree resolution) is used for the analysis of sea ice data, while you state earlier that the model has a ~1° resolution in the ocean and sea?

> **Reply**: In the CESM1, the sea-ice model is coupled more "tightly" to the atmosphere and land models than to the ocean model, in order to better resolve the diurnal cycles (Craig et al., 2012). As we focus on long-term variability and trends, the spatial resolution for the analysis is not that important as detailed local variability is not the main interest here.
>
> Craig, A. P., Vertenstein, M., and Jacob, R.: A new flexible coupler for earth system modeling developed for CCSM4 and CESM1, Int. J. High Perform. Comput. Appl., 26, 31–42, https://doi.org/10.1177/1094342011428141, 2012.

Line 191: Some details on wavelet analysis are missing. Have you detrended the records prior to the wavelet analysis? Were the series resampled (I assume so), and to which common time increment? How the autoregression in the series was estimated prior to significance testing?

> **Reply**: During the analysis, the data was internally detrended. For that, a degree of time series smoothing of 0.75 was used over 100 simulations. For the surrogate time series, we used an autoregressive model (AR-1). The time resolution for the model data was 1 year and the time resolution for the proxy data varied among the records. We added more details about all the settings used in the wavelet analysis (lines 199-204 in the marked up version). The time resolutions for each proxy record were added to Figures S8 to S14 in the Supplementary Material.

Line 195: What was the point in log-transforming the data prior to analysis?

**Reply**: It is common practice to log-transform fractional data for statistical analysis to avoid the sharp cut-off at 0 and 1 that does not exist for the other variables we compare to (both proxies and simulated temperature etc.). Log-transformation helps to normalize the data distribution and reduces skewness and heteroscedasticity to make it comparable with non-fractional data (lines 208-210 in the marked up version).

Line 208: Diatoms; as an option MD99-2322 series with a relatively good resolution: Miettinen, et al., doi:10.1002/2015PA002849.

**Reply**: We looked at MD99-2322 data, but only the "sea ice concentration" data is available, and not the abundance of diatom indicator species. To avoid the input of more bias sources, we decided not to use quantitative estimates of reconstructed sea-ice concentrations and sea-ice duration.

Line 223: "…group refer to increases in sea-ice cover." As the authors notice later, for IP25 with its unimodal response to sea ice, the presence of perennial ice cover leads to the opposite tendency.

**Reply**: Indeed, that is an issue related to the interpretation of $IP_{25}$ data. However, because most of the sediment cores were not retrieved in regions with perennial ice cover, the unimodal response of $IP_{25}$ to sea ice was kept only in the discussion (lines 356-357 in the marked up version).

Line 284: it is not a "binary behavior", rather a manifestation of a unimodal response model quite typical for many climate proxies. See e.g. in Oksman et al., (2019) in Section 2, or elsewhere on quantitative methods of paleoreconstructions.

**Reply**: The term "binary behaviour" was replaced by "two end-member scenarios for absent $IP_{25}$", as used in Belt (2018).

While it is true that microfossil species typically have unimodal responses to environmental variables (optimum and tolerance), increases in sediment $IP_{25}$ concentration is generally associated with seasonal sea ice but not quantitatively. Moreover, the absence of $IP_{25}$ can indicate either absence of sea ice or perennial sea ice, described with "two end-member scenarios" or "binary" (Belt, 2018, 2019). Thus, we do not think unimodal is the correct response model here, but we changed "binary" to "two end-member scenarios for absent $IP_{25}$" (lines 325 and 357 in the marked up version).

Belt, S. T.: Source-specific biomarkers as proxies for Arctic and Antarctic sea ice, Org. Geochem., 125, 277–298, https://doi.org/10.1016/j.orggeochem.2018.10.002, 2018.

Belt, S. T.: What do IP25 and related biomarkers really reveal about sea ice change?, Quat. Sci. Rev., 204, 216–219, https://doi.org/10.1016/j.quascirev.2018.11.025, 2019.

Line 287: "… switch from long seasonal sea-ice cover to a multiyear sea-ice scenario" i.e. transition to a perennial sea ice cover in the area

> **Reply**: The sentence was fixed as suggested (line 328 in the marked up version).

Line 304 "…Therefore, it does not have enough temporal coverage to register a potential IP25 decrease. "

Or this could actually be due to a more southerly location of the record. This is a very dynamic area close to the oceanic front, and spatial surface gradients are large.

> **Reply**: Unfortunately, this line of discussion does not work because core C (MSM05/5_723-2) was retrieved from a slightly northern location than core B (MSM05/5_712-1). Figure 3 was fixed to avoid this misinterpretation.

Line 309 see my earlier comment, this is not a "binary behaviour"

> **Reply**: As mentioned earlier, the term "binary behaviour" was replaced by "two end-member scenarios for absent $IP_{25}$" (lines 325 and 357 in the marked up version).

Line 324: "One possible explanation for this difference is the impact of warm waters on the melting rate of the drift ice". What is actually implied here? **Stronger oceanic fronts** in the area? Higher sea ice export to the area for this particular time period?

> **Reply**: We are referring to the presence of stronger oceanic fronts. More specifically, the presence of stronger currents carrying warm waters in the Denmark Strait between 1450 and 1650 CE. Because sediment cores JR51-GC35 (record F) and MD99-2269 (record G) were collected in regions relatively more protected than sediment core MD99-2263 (record H), they were not as influenced by this intrusion of warm waters as the area of the sediment core MD99-2263 (record H). This sentence was rephrased to improve clarity (lines 372-374 in the marked up version).

Line 393: "mean chunks" consider changing to "data segments"..

> **Reply**: The term "data segments" would be quite unspecific. "Mean chunks" are meant to highlight that non-overlapping mean segments are used mimicking the samples of proxy records. We defined the expression in more detail when first using it (lines 193-194 in the marked up version).

Line 394: Refs to figures in Supplementary. Remarkable that CESM1 EOF1 shows very little loading south of Greenland and in the Labrador sea compared to MPI. What could be the reason for this, lack of (sea ice) related variability in the region, or it just went into another, presumably second EOF?

**Reply**: Since it's about the area south of Greenland, we assume this question is about the winter season (year maximum area). We checked, and the sea-ice variability was not "transferred", or better captured by the second EOF (see figure attached – "EOF1_2_CESM_MPI_winter.pdf"). The apparent lack of sea-ice variability on southern Labrador Sea in CESM1 might have been caused by two reasons. One is an artifact of the plots' scales: EOF1 ranges from -0.3 to +0.3 in CESM1, but from -0.2 to +0.2 in MPI. Another possible explanation is that the sea-ice variability in the Greenland Sea is simply stronger than in the southern Labrador Sea in the CESM1 than in MPI model. As we focus on the large-scale variability patterns, inter-model differences in local variability will have little impact on long-term variations.

[Figure]

Line 411: "close to Greenland east coastline" better change to the "Greenland sea", as "close to east coast" can be interpreted ambiguously

> Reply: It was corrected as suggested (line 472 in the marked up version).

Line 416: "...which affects...that reaches Greenland sea. " Not clear what the authors meant here. I suggest a full stop after the "the Fram Strait" an leave out the rest of the sentence.

> Reply: It was corrected as suggested (lines 477-478 in the marked up version).

Line 419: "..PC1 in both models contains some cyclicity...". Better use the term "quasi periodic variability" for this particular case, not "cyclicity". Same in line 421

> Reply: It was corrected as suggested (lines 481-483 in the marked up version).

Line 455  "…the variability was mostly concentrated in periods around 30 and 50 years."

Since some information on wavelet analysis is missing, my thoughts below can be a bit speculative, but good to check it anyways. This "statistically significant" variability emerges only for some shorter periods when the data sampling is high enough to resolve any changes at these time scales. For significance testing, most likely an AR1 model was used based on autocorrelation coefficient estimated from the data directly. Due to data resampling/interpolation the autocorrelation will be overestimated, hence leading to lower CI 95% ranges at shorter timescales. With such background model for the analyzed time series, sporadically emerging variability at shorter time scales will be very much likely identified as significant anyways, yet being an artifact of testing procedure.

I therefore would suggest the authors to consider how meaningful is to make any numerical comparisons for the sub-centennial timescales of variability between the proxies and models since a temporal resolution of the proxy based reconstructions for most of the records is fairly low.

> Reply: We agree that sub-centennial variability may not be very reliable in proxy data which we mention two sentences later and again in line 508 below (now lines 522-526, and 580-583 in the marked up version). The question to which extent it is meaningful is part of why we included it in comparison to climate models, albeit only as a side topic with supplementary figures.
>
> As part of explaining the details of the wavelet analysis in more detail in the revised version (lines 199-204 in the marked up version), we added an additional sentence on the issue. We interpolated all proxy records onto a regular timescale using a linear interpolation before performing the wavelet analysis. So, the wavelet does not directly "know" about data sampling issues but indirectly suffers from the overestimated persistence from the interpolation. Obviously, as you correctly assume, the AR-1 test will also suffer from an overestimated autocorrelation due to the interpolation procedure. This is a common problem with all kinds of proxy data. The deteriorated signal-to-noise-ratio in combination with overestimated persistence would, however, rather underestimate significance and can be assumed to be on the conservative side. As we do not focus on sub-centennial variability in proxies in the main text, we think it is acceptable to keep the significance test as is in the

supplementary figures. But we expanded the explanation of the issue in the main text (lines 526-532 in the marked up version).

Line 508. "…variability and noise in the proxy data have a marked impact on short-them variability" Definitely in this case the lack of temporal resolution in the proxy series as well as dating uncertainties should be mentioned.

**Reply**: Indeed. Thus, we expanded the explanation as written above (lines 526-532 in the marked up version). We also mentioned in the conclusion (lines 580-582 in the marked up version) the importance of poor temporal resolution and dating uncertainties on the interpretation of short-them variability which is why we focus mainly on low-frequent variations in this study.

---

## Referee Report (RR1)

The authors adequately addressed the comments/suggestion made to the original version of the manuscript. Please find below a few more generally minor comments I would suggest the authors to consider before the draft can be published. The line numbers refer to the new version of the document with changes highlighted.

Line 300:

"Interestingly, G1 demonstrates a higher rate of sea-ice increase from 1600 CE until the LIA, which is not apparent in the general trend (Figure 2)."

Period of 1600 until late 1800s is associated with LIA, so the statement "from 1600CE until the LIA" needs to be corrected/rephrased. Otherwise, it appears from this statement that LIA is associated with a specific time point, not a period.

Line 423:

"The satellite data, on the other hand, is the sea-ice extent, which considers the sum of the area of all grid cells covered by at least 15% of sea ice."

Why did the authors then used the extent, while the area is also available directly from the NSIDC SSMI data? This would make the comparison way more consistent. For winter the difference between the two metrics will not be very substantial, but in the summer this is not the case. I recommend the authors to switch to similar variables.

Line 523:

"This difference is likely caused by internal climate and ecosystem dynamics where proxy-based sea-ice reconstructions were affected by complex environmental parameters, while the physical models directly responded to large-scale atmospheric forcing from variations in solar, volcanic, and orbital forcing. Furthermore, while the proxy records used here have relatively high resolution (< 100 years), the resolution is generally not enough to capture multidecadal oscillations."

In addition and in my personal opinion, are likely most important, chronology errors of various nature (core sub-sampling itself, delta R uncertainty, depth age modelling method and the associated uncertainty etc) is also a serious obstacle when comparing various oceanic proxy series on sub-centennial time scales.

Line 531

"dominance" appears twice.

Line 531:

"…while periods with high resolution confirm the dominance of dominance of multidecadal variability …"

The authors can easily estimate how correct this statement is directly from wavelet analysis by summing the variance over the band of timescales corresponding to multidecadal variability. See example in Torrence and Compo, (1998).
It is also useful to indicate (again) that this dominance applies to the timescales over a decade, since the analysis is applied to filtered series with sub-decadal variations removed. Otherwise, annual to intra-annual variations generally contribute most to the total variance.

Line 535:
"The spatial distribution of EOF2, associated with the multidecadal to centennial-scale fluctuations…"

Better, in my opinion, refer instead to a "Spatial structure of EOF2" or a "Loading pattern"

Line 537:

"This inconsistency is probably caused by the low explained variance of PC2 (15% for CESM1 and 5% for MPI-ESM), incorporating a large degree of residual noise. Although the multidecadal and centennial variability of Arctic summer sea ice has been linked to changes in the northward Atlantic and Pacific heat transport and in the Arctic dipole pattern, there is still some significant variability between mean states of Arctic sea ice simulated by different models (Li et al., 2018)."

Another possible explanation is purely methodological, namely the orthogonality, by definition/construction, of EOFs in the basic EOF analysis. It means, in plain words, that EOF1 has a unipole spatial pattern, EOF2 - bi-pole, EOF3 4-pole etc. For the region used sea ice is mainly present on both sides of Greenalnd, this is where EOF2 will form its two poles with loadings of the opposite sign.

Line 577:

"The recent sea-ice retreat, on the other hand, was likely initiated by a recovery from the volcanic dust emissions…"

Better rephrase to "recovery after the LIA", since volcanism might have triggered a number of feedbacks that caused lasting negative SAT and positive sea ice anomalies that are now associated with the LIA manifestation. From the way it is written now, an impression of a direct liner response to continuous volcanic dust emission emerges.

And the final comment concerns the hypothesis testing in wavelet power spectra (Starting from line 527). In their response letter the authors agree that resampling to the annual scale causes unrealistically high values of the AR-1 coefficient used in hypothesis testing. The authors, however, are not correct stating in the response that "The deteriorated signal-to-noise-ratio in combination with overestimated persistence would, however, rather underestimate significance and can be assumed to be on the conservative side." Due to a redistribution of variance towards lower frequencies in the AR1 series with a very high autocorrelation coefficient (very close to 1), the significance threshold for hypothesis testing will be underestimated in the higher frequency range, and overestimated in the lower frequency range. The actual configuration will depend on the AR(1) value of the background process that could generate the observed series. I therefore consider that the use of thresholds for significance testing calculated from the resampled data is not correct. Instead, I recommend the authors (if they would like to retain the discussion that involves significance testing from wavelet spectra of the proxy series) to use the AR(1) coefficient estimated from the original unevenly sampled data. Since the authors used the R environment in their calculations, it should not be a problem to use RedFit method (Schulz and Mudelsee, 2002) for this purpose. The implementation of RedFit can be found in dplR package (https://cran.r-project.org/web/packages/dplR/dplR.pdf ). Once the AR(1) coefficients estimates are obtained, you can disable the automatic calculation of AR1 in wavelet power spectra computation/analysis

procedure and type in the RedFit estimated coefficients instead. This will provide you a way more fair view on the wavelet spectra of proxy series.

References:

Schulz, M. and Mudelsee, M. (2002) REDFIT: estimating red-noise spectra directly from unevenly spaced paleoclimatic time series. Computers & Geosciences, 28(3), 421–426.

Torrence, C., and G. P. Compo, 1998: A Practical Guide to Wavelet Analysis. *Bull. Amer. Meteor. Soc.*, **79**, 61–78, https://doi.org/10.1175

---

## Author Response (AR2)

**Editor decision: Publish subject to minor revisions (review by editor)**

by Sebastian Gerland, 10 Jan 2024

Public justification (visible to the public if the article is accepted and published):

Dear Ana Lucia Lindroth Dauner,

thank you for the revision of your manuscript "Sea ice variations and trends during the Common Era in the Atlantic sector of the Arctic Ocean". With the changes you made, in the next step only minor revisions would be necessary. Please see the new comments by the two reviewers and take them into account in your new revision.

Best regards

Sebastian Gerland

> **Reply**: We appreciate the additional comments and took them into account in this latest version. The lines refer to the marked-up version.

**Report #1 - Anonymous referee #1, Submitted on 20 Dec 2023**

**Summary:**

Dear author team,

thank you for taking into account the review comments provided. I only found a few technical / minor things that you might want to correct and/or comment on.

**Specific comments:**

L40 / L43 (as an example, please check the entire manuscript): I find "palaeo" and "paleo" ... what is correct?

> **Reply**: We kept "palaeo" to keep consistent with the British spelling (lines 43, 95 and 553 in the marked-up version).

L110: You explained the meaning of IP25 already in L78; perhaps it can be deleted here?

> **Reply**: Done as suggested (line 110 in the marked-up version).

L330: While you give Smith and Barber as a reference here I was wondering whether this statement really holds the way as written. The two main polynyas around Greenland at the NOW = the North Open Water polynya which forms regularly southwest of Nares Strait and the NEW = the North-East

Water polynya close to the Fram Strait where grounded icebergs in combination with perennial sea ice block the everlasting southward ice export through Fram Strait. These are just two comparably small polynya areas - in contrast to, e.g. the Eastern Antarctic where polynyas are really abundant, or the Arctic flaw lead / polynya system. Given the forcing conditions on Greenland's eastern side I suggest to stress here that the polynyas you hypothesize to have formed in the past all formed along Greenland's western side.

> **Reply**: Done as suggested (line 330 in the marked-up version).

L412-418: If you would have used sea ice area data (from satellite observations) like available from https://www.cen.uni-hamburg.de/icdc or from https://met.no then you would have been able to make a clean 1-to-1 comparisons instead of rambling about one is area but the other is extent. This is a bit sub-optimal.

> **Reply**: We have now used the sea ice area data from https://www.cen.uni-hamburg.de/icdc to compare the numerical model results to the satellite data (lines 418 – 424 in the marked-up version).

L555: Looking back at the discussions provided, I note that you did not take into account any potential changes in the snow cover on top of the sea ice. Snow also influences light availability under the sea ice quite a bit. Possibly you have a reason for this?

> **Reply**: We have added snowpack thickness into the discussion (lines 553 – 555 in the marked-up version).

**Report #2 - Referee #2: Dmitry Divine (dima@npolar.no), Submitted on 10 Jan 2024**

**Summary:**

The authors adequately addressed the comments/suggestion made to the original version of the manuscript. Please find below a few more generally minor comments I would suggest the authors to consider before the draft can be published. The line numbers refer to the new version of the document with changes highlighted.

**Comments:**

Line 300: "Interestingly, G1 demonstrates a higher rate of sea-ice increase from 1600 CE until the LIA, which is not apparent in the general trend (Figure 2)."

Period of 1600 until late 1800s is associated with LIA, so the statement "from 1600CE until the LIA" needs to be corrected/rephrased. Otherwise, it appears from this statement that LIA is associated with a specific time point, not a period.

> **Reply**: We rephrased the sentence as suggested (line 295 in the marked-up version).

Line 423: "The satellite data, on the other hand, is the sea-ice extent, which considers the sum of the area of all grid cells covered by at least 15% of sea ice."

Why did the authors then used the extent, while the area is also available directly from the NSIDC SSMI data? This would make the comparison way more consistent. For winter the difference between the two metrics will not be very substantial, but in the summer this is not the case. I recommend the authors to switch to similar variables.

> **Reply**: As suggested by the other referee, we now have used the sea ice area data from https://www.cen.uni-hamburg.de/icdc to compare the numerical model results to the satellite data (lines 418 – 424 in the marked-up version).

Line 523: "This difference is likely caused by internal climate and ecosystem dynamics where proxy-based sea-ice reconstructions were affected by complex environmental parameters, while the physical models directly responded to large-scale atmospheric forcing from variations in solar, volcanic, and orbital forcing. Furthermore, while the proxy records used here have relatively high resolution (< 100 years), the resolution is generally not enough to capture multidecadal oscillations."

In addition and in my personal opinion, are likely most important, chronology errors of various nature (core sub-sampling itself, delta R uncertainty, depth age modelling method and the associated uncertainty etc) is also a serious obstacle when comparing various oceanic proxy series on subcentennial time scales.

> **Reply**: We have now addressed the chronology errors in our discussion (lines 525 – 528 in the marked-up version).

Line 531: "dominance" appears twice.

> **Reply**: The repeated word was removed (line 532 in the marked-up version).

Line 531: "…while periods with high resolution confirm the dominance of dominance of multidecadal variability …"

The authors can easily estimate how correct this statement is directly from wavelet analysis by summing the variance over the band of timescales corresponding to multidecadal variability. See example in Torrence and Compo, (1998). It is also useful to indicate (again) that this dominance applies to the timescales over a decade, since the analysis is applied to filtered series with sub-decadal variations removed. Otherwise, annual to intra-annual variations generally contribute most to the total variance.

> **Reply**: This sentence refers to the wavelet analysis based on the proxy records, which were not filtered to remove the sub-decadal variability. Most of the sediment core resolutions don't allow the analysis of sub-decadal variations. But we reinforced that the model data for the wavelet analysis was filtered to remove the sub-decadal variations. And the variance over the band of timescales is represented in panels C in the Supplementary Figures S8 to S14. The

periods marked with the red dots in those panels represent the same as the periods above "the upper dashed line is the 95% confidence spectrum" in Torrence and Compo, (1998).

Line 535: "The spatial distribution of EOF2, associated with the multidecadal to centennial-scale fluctuations…"

Better, in my opinion, refer instead to a "Spatial structure of EOF2" or a "Loading pattern".

> **Reply**: Changed as suggested (lines 475 and 537 in the marked-up version).

Line 537: "This inconsistency is probably caused by the low explained variance of PC2 (15% for CESM1 and 5% for MPI-ESM), incorporating a large degree of residual noise. Although the multidecadal and centennial variability of Arctic summer sea ice has been linked to changes in the northward Atlantic and Pacific heat transport and in the Arctic dipole pattern, there is still some significant variability between mean states of Arctic sea ice simulated by different models (Li et al., 2018)."

Another possible explanation is purely methodological, namely the orthogonality, by definition/construction, of EOFs in the basic EOF analysis. It means, in plain words, that EOF1 has a unipole spatial pattern, EOF2 - bi-pole, EOF3 4-pole etc. For the region used sea ice is mainly present on both sides of Greenland, this is where EOF2 will form its two poles with loadings of the opposite sign.

> **Reply**: This explanation was added in our discussion (lines 539 – 541 in the marked-up version).

Line 577: "The recent sea-ice retreat, on the other hand, was likely initiated by a recovery from the volcanic dust emissions…"

Better rephrase to "recovery after the LIA", since volcanism might have triggered a number of feedbacks that caused lasting negative SAT and positive sea ice anomalies that are now associated with the LIA manifestation. From the way it is written now, an impression of a direct liner response to continuous volcanic dust emission emerges.

> **Reply**: This sentence was changed as suggested (line 582 in the marked-up version).

And the final comment concerns the hypothesis testing in wavelet power spectra (Starting from line 527). In their response letter the authors agree that resampling to the annual scale causes unrealistically high values of the AR-1 coefficient used in hypothesis testing. The authors, however, are not correct stating in the response that "The deteriorated signal-to-noise-ratio in combination with overestimated persistence would, however, rather underestimate significance and can be assumed to be on the conservative side." Due to a redistribution of variance towards lower frequencies in the AR1 series with a very high autocorrelation coefficient (very close to 1), the significance threshold for hypothesis testing will be underestimated in the higher frequency range, and overestimated in the lower frequency range. The actual configuration will depend on the AR(1) value of the background process that could generate the observed series. I therefore consider that

the use of thresholds for significance testing calculated from the resampled data is not correct. Instead, I recommend the authors (if they would like to retain the discussion that involves significance testing from wavelet spectra of the proxy series) to use the AR(1) coefficient estimated from the original unevenly sampled data. Since the authors used the R environment in their calculations, it should not be a problem to use RedFit method (Schulz and Mudelsee, 2002) for this purpose. The implementation of RedFit can be found in dplR package (https://cran.rproject.org/web/packages/dplR/dplR.pdf ). Once the AR(1) coefficients estimates are obtained, you can disable the automatic calculation of AR1 in wavelet power spectra computation/analysis procedure and type in the RedFit estimated coefficients instead. This will provide you a way more fair view on the wavelet spectra of proxy series.

Reply: We tested the RedFit method using the dplR package for a few records, and re-run the wavelet analysis using the coefficient estimated by the Redfit function, to compare with the default AR(1). As you can see below, there is not much difference between using the default AR(1) coefficient and the coefficient estimated for the original unevenly sampled data (RedFit function).

Because there are no significant differences and because we do not discuss in detail the significant periods observed in each proxy record, we consider that using the automatic calculation of AR(1) coefficient is enough for our analysis.

[Figure]

| Wavelet using default AR(1) | Wavelet using RedFit coefficient |
|---|---|

[Figure]

| Wavelet using default AR(1) | Wavelet using RedFit coefficient |
|---|---|

[Figure]

| Wavelet using default AR(1) | Wavelet using RedFit coefficient |
|---|---|

[Figure]

| Wavelet using default AR(1) | Wavelet using RedFit coefficient |
|---|---|

[Figure]

**References:**

Schulz, M. and Mudelsee, M. (2002) REDFIT: estimating red-noise spectra directly from unevenly spaced paleoclimatic time series. Computers & Geosciences, 28(3), 421–426.

Torrence, C., and G. P. Compo, 1998: A Practical Guide to Wavelet Analysis. Bull. Amer. Meteor. Soc., 79, 61–78, https://doi.org/10.1175

---

## Author Response (AR3)

**Editor decision: Publish subject to minor revisions (review by editor)**

by Sebastian Gerland, 13 Feb 2024

Public justification (visible to the public if the article is accepted and published):

Dear Ana Lúcia Lindroth Dauner,

thank you for the newly revised version of your manuscript, along with explanations and version with tracked changes. This is very much appreciated.

I found only a few remaining points I would like you to consider following up, see list below. The line numbers used refer to line numbers in the ATC3 file.

Thank you very much for your work with this.

Best regards

Sebastian Gerland

> **Reply**: We appreciate the additional comments and made the needed alterations in this latest version. The lines refer to the marked-up version.

Your response to the comments of reviewer #1:

• Lines 553-555: I suggest to support the statement on the role of snow with a reference (and consider refining the statement according to which reference is used). See for example in Leu et al. 2015 (Progress in Oceanography 139 (2015) 151–170).

> **Reply**: We improved this sentence and added suitable references (line 546-550 in the marked-up version).

Your response to the comments of reviewer #2:

• Line 295: I suggest to at "the" ahead of "LIA".

> **Reply**: Done as suggested (line 295 in the marked-up version).

• Line 475: Assuming this was a typo: Remove "e" in "estructure".

> **Reply**: Indeed, it was a typo. Done as requested (line 469 in the marked-up version).

• Line 540: remove double space (between "EOF" and "will") and change "." to ";" after "pattern", or rephrase last sentence ("In this case …") so it would become a complete sentence.

> **Reply**: Done as requested (line 533 in the marked-up version).

Other comments:

• Lines 420-424: When using 10exp6 instead of «million», please add a dot ahead of 10exp6. However, I suggest to use instead "million", as for example also done in lines 427 and 442.

> **Reply**: We replaced the expression "10exp6" for "million" (lines 414-418 in the marked-up version).

• Lines 440-443, and figure caption of Figs. S8-S14: There should be no space between number and "%".

> **Reply**: Done as requested throughout the entire manuscript.

• The sector shown as modelled area (red lines) in the map in graphical abstract reappears in Figures 3b, 6cdgh, S6 and S7. However, I could not find an explanation of this in any of the figure captions. I suggest to add an explanation where it first appears (Fig. 3b), beyond the graphical abstract. Else, I see that the modelled area is drawn in the south slightly beyond the 90 west and 30 east meridians, I wonder if that is done on purpose (if the modelled area is really like this), or if this is a drawing inaccuracy. If the latter is the case I suggest to redraw the area in the actual figures.

> **Reply**: We added the explanation of the thick black line in the caption of Figure 3 as requested. We also fixed the drawing of the line to coincide with the 90 west and 30 east meridians in all figures.